# aCPSF1 cooperates with terminator U-tract to dictate archaeal transcription termination efficacy

**Jie Li[1][*][†], Lei Yue[1,2][†], Zhihua Li[1,2][†], Wenting Zhang[1,2], Bing Zhang[2,3], Fangqing Zhao[2,3], Xiuzhu Dong[1,2][*]**

[1]State Key Laboratory of Microbial Resources, Institute of Microbiology, Chinese Academy of Sciences, Beijing, China; [2]University of Chinese Academy of Sciences, Beijing, China; [3]Beijing Institutes of Life Science, Chinese Academy of Sciences, Beijing, China

**\*For correspondence:**
lijie824@im.ac.cn (JL);
dongxz@im.ac.cn (XD)

[†]These authors contributed equally to this work

**Competing interest:** The authors declare that no competing interests exist.

**Abstract** Recently, aCPSF1 was reported to function as the long-sought global transcription termination factor of archaea; however, the working mechanism remains elusive. This work, through analyzing transcript-3′end-sequencing data of *Methanococcus maripaludis*, found genome-wide positive correlations of both the terminator uridine(U)-tract and aCPSF1 with hierarchical transcription termination efficacies (TTEs). In vitro assays determined that aCPSF1 specifically binds to the terminator U-tract with U-tract number-related binding affinity, and in vivo assays demonstrated the two elements are indispensable in dictating high TTEs, revealing that aCPSF1 and the terminator U-tract cooperatively determine high TTEs. The N-terminal KH domains equip aCPSF1 with specific-binding capacity to terminator U-tract and the aCPSF1-terminator U-tract cooperation; while the nuclease activity of aCPSF1 was also required for TTEs. aCPSF1 also guarantees the terminations of transcripts with weak intrinsic terminator signals. aCPSF1 orthologs from Lokiarchaeota and Thaumarchaeota exhibited similar U-tract cooperation in dictating TTEs. Therefore, aCPSF1 and the intrinsic U-rich terminator could work in a noteworthy two-in-one termination mode in archaea, which may be widely employed by archaeal phyla; using one trans-action factor to recognize U-rich terminator signal and cleave transcript 3′-end, the archaeal aCPSF1-dependent transcription termination may represent a simplified archetypal mode of the eukaryotic RNA polymerase II termination machinery.

## Editor's evaluation

The process of termination in Archael species is poorly defined despite a high relation to eukaryotes and a shared homology of termination factors. In this study, the authors defined key features that drive termination to include an upstream uridine track that is bound by the CPSF ribonuclease through KH RNA binding domains not present in the CSPF counterparts. This work provides fundamental mechanistic insight into the conserved manner of termination in the archael species.

## Introduction

Transcription termination is an essential and highly regulated process in all forms of life, which not only determines the accurate 3′-end boundary of a transcript and transcription-related regulatory events, but is also important in shaping programmed transcriptomes of living organisms (*Peters et al., 2011*; *Porrua et al., 2016*; *Porrua and Libri, 2015*; *Ray-Soni et al., 2016*; *Yue et al., 2020*). Highly controlled transcription termination, which prevents read-through resulted undesired increases in downstream

coding regions and the accumulation of antisense transcripts, can be particularly important in prokaryotes because of their densely packed genomes (*Peters et al., 2012*; *Yue et al., 2020*).

Research has indicated that bacteria primarily employ two transcription termination mechanisms, Rho-dependent and -independent (intrinsic). In the Rho-dependent mechanism, the RNA translocase Rho, via recognizing a cytosine-rich sequence in nascent transcripts, dissociates the processive transcription elongation complex (TEC) based on its ATPase activity. In contrast, the intrinsic termination merely depends on a nascent RNA structure with a 7–8 base-paired hairpin followed by a run of uridines (Us) (*Gusarov and Nudler, 1999*; *Peters et al., 2011*; *Porrua et al., 2016*; *Ray-Soni et al., 2016*). A bacteria-like intrinsic termination mechanism that depends on a U-stretch is also found in the eukaryotic RNA polymerase (RNAP) III (*Nielsen et al., 2013*; *Orioli et al., 2011*). Distinctively, transcription termination of the eukaryotic RNAP II, which transcribes mRNAs and non-coding RNAs, usually involves a transcript 3′-end processing event, in which the cleavage and polyadenylation factor complex (CPF/CPSF), under assistance of the accessory cleavage factors CFIA and CFIB, recognizes the termination signal, a poly(A) site at transcript 3′-end. Following recognition, the CPF/CPSF complex cleaves downstream the termination signal of the nascent RNA and polyadenylates at the cleaved 3′-end for mRNA maturation, and triggers RNAP II dissociation for transcription termination (*Baejen et al., 2017*; *Eaton et al., 2018*; *Grzechnik et al., 2015*; *Kim et al., 2004*; *Kuehner et al., 2011*; *Larochelle et al., 2018*; *Porrua et al., 2016*).

Compared with bacteria and eukaryotes, knowledge of the transcription termination mechanisms in the third form of life, archaea, is very limited (*Dar et al., 2016a*; *Maier and Marchfelder, 2019*). Archaea represent a primary domain of cellular life and phylogenetically are more closely related to eukaryotes than bacteria (*Eme et al., 2017*; *Williams et al., 2020*; *Zaremba-Niedzwiedzka et al., 2017*). Specifically, archaea employ a eukaryotic RNAP II homolog, archaeal RNAP (aRNAP) (*Werner and Grohmann, 2011*), but have compact genomes with short intergenic regions (IGRs) and co-transcribed polycistronic operons, highlighting the importance of a controllable transcription termination. Earlier studies have suggested that, similar to the bacterial intrinsic termination, transcription termination of aRNAP may depend on a short U-rich sequence at the transcript 3′-end but with no strict requirements of an upstream hairpin structure (*Hirtreiter et al., 2010*; *Maier and Marchfelder, 2019*; *Santangelo et al., 2009*; *Santangelo and Reeve, 2006*; *Spitalny and Thomm, 2008*; *Thomm et al., 1993*). Recently, Term-seq, an approach that enables accurate mapping of all exposed RNA 3′-ends in prokaryotes and determines the transcription termination sites (TTSs) at the genome-wide level in representative bacteria and archaea (*Dar et al., 2016b*; *Porrua et al., 2016*; *Yue et al., 2020*), has been developed. Through Term-seq, U-rich sequences preceding TTSs, without preceding hairpin structures, were identified to be overrepresented in the transcripts of four representative archaeal species: *Methanosarcina mazei*, *Sulfolobus acidocaldarius*, *Haloferax volcanii*, and *Methanococcus maripaludis* (*Berkemer et al., 2020*; *Dar et al., 2016b*; *Yue et al., 2020*). Therefore, the U-tract sequences at the transcript 3′-ends are assumed to be the intrinsic termination signals of archaea; in addition, without strictly requiring an upstream hairpin structure in most of the archaeal terminator sequences suggests a distinct intrinsic termination mechanism of archaea from that of bacteria (*Maier and Marchfelder, 2019*).

The protein factors that mediate archaeal transcription termination have been reported in recent years. The *Thermococcus kodakarensis* Eta (*eu*archaeal *t*ermination *a*ctivity) has been reported to transiently engage the TEC and release the stalled TEC from damaged DNA lesions, resembling the bacterial Mfd termination factor and functioning specifically in response to DNA damage (*Walker et al., 2017*). Most recently, aCPSF1, also named FttA (*F*actor that *t*erminates *t*ranscription in *A*rchaea), has been demonstrated as a transcription termination factor of archaea because it could competitively disrupt the processive TEC at normal transcription elongation rate and implement a kinetically competitive termination dependent on both the stalk domain of RNAP and the transcription elongation factor Spt4/5 in vitro (*Sanders et al., 2020*). aCPSF1 is affiliated within the β-CASP ribonuclease family, and is ubiquitously distributed in all archaeal phyla (*Li et al., 2021*; *Phung et al., 2013*; *Yue et al., 2020*). Initially, aCPSF1 was assumed to function in RNA maturation and turnover of Archaea (*Clouet-d'Orval et al., 2015*), and endoribonuclease activity was identified for three aCPSF1 orthologs in vitro (*Levy et al., 2011*; *Phung et al., 2013*; *Silva et al., 2011*), with one also exhibiting 5′–3′ exoribonuclease activity (*Phung et al., 2013*). Our recent study reported that aCPSF1, depending on its ribonuclease activity, controls in vivo transcription termination at the genome-wide level and

ensures programmed transcriptome in *M. maripaludis*, and its orthologs from the distant relatives, *Lokiarchaeota* and *Thaumarchaeota*, implement the same function in termination (*Yue et al., 2020*). However, although the in vitro enzymatic assay determined that aCPSF1 primarily and endoribonucleolytically cleaves downstream of a U-rich motif that precedes the identified TTSs (*Yue et al., 2020*), some open questions remain, such as (i) whether the aCPSF1-dependent and the U-tract terminator-based intrinsic terminations are two independent mechanisms, or the two in fact work cooperatively in archaea, or (ii) if the aCPSF1-dependent termination simply serves as a backup mechanism for the genes/operons containing less-efficient intrinsic termination signals as assumed (*Sanders et al., 2020*; *Wenck and Santangelo, 2020*); (iii) what the exact sequence motifs that aCPSF1 recognizes are, and (iv) whether, like the eukaryotic multiple subunit composed termination complex, aCPSF1 also requires others to recognize the termination signals.

In the present work, via an intensive analysis of the Term-seq data in *M. maripaludis,* we comprehensively evaluated the correlations of the transcription termination efficacies (TTEs) for all identified TTSs with both the cis-element U-tract terminator and the trans-action termination factor aCPSF1. Further, in combination with molecular and genetic validations, we determined that aCPSF1 and the terminator U-tract cooperatively dictate high TTEs. The in vitro and in vivo assays together demonstrated that the N-terminal K homolog (KH) domains of aCPSF1 specifically recognize and bind to the terminator U-tract. Therefore, the archaeal termination factor aCPSF1 could accomplish the U-tract terminator recognition and transcript 3′-end cleavage by itself, and the factor-dependent transcription termination may be the primary mechanism used by archaea.

## Results
### A positive correlation is found between the TTEs and the terminator four-uridine (U4) tract numbers preceding TTSs in *M. maripaludis*

In an attempt to evaluate the specific termination signals recognized by the termination factor aCPSF1 and its role in dictating the in vivo TTEs, we intensively reanalyzed the Term-seq data of *M. maripaludis* obtained previously (*Yue et al., 2020*). By following a stringent filtration workflow in TTS definition and to preclude identifying sites derived from stale RNA processing or degradation products, TTS searching was restricted within 200 nt downstream of the stop codon of a gene. This served to maximally enrich the authentic TTSs near the gene 3′-ends and only sites that appeared in both biological replicates with high coverage (see Materials and methods) were selected. In total, 2357 TTSs were obtained, including the previously identified 998 primary and 1,359 newly identified secondary TTSs (*Supplementary file 1*). Multiple consecutive TTSs were found in >50% of transcription units (TUs) of *M. maripaludis* (*Figure 1—figure supplement 1*), which could produce multi-isoforms of a transcript with varying 3′-UTRs, as found in *M. mazei* and *S. acidocaldarius* (*Dar et al., 2016a*). Nevertheless, compared with the primary TTSs, which have the highest Term-seq reads among all identified 3′-end sites in each TU, much lower median read abundances, TTEs, and motif scores were found for the secondary TTSs (*Figure 1—figure supplement 2*). This indicates that TUs are mainly terminated at the primary TTSs, which were therefore used for further investigation.

Sequence analysis of the 961 primary TTSs of coding TUs found a featured terminator motif, a 23 nt U-tract with four consecutive uridine nucleotides (U4) that are most proximal to the TTS having the highest matching. To evaluate the contribution of the U-tract sequence preceding TTSs to transcription termination in *M. maripaludis*, we first defined and calculated TTE of each TU. After inspecting the genome-wide Term-seq mapping file, a dramatic decreasing pattern was observed in the mapping reads at four nucleotides (nts) between sites +2 and −2 flanking TTS (−1 nt) in the majority of the primary TTSs (*Figure 1A* and *Figure 1—figure supplement 3*). This indicates that transcription appears to be terminated most frequently at the four nucleotides, which was therefore defined as the TTS quadruplet. Through pair-wisely comparing the reads of each nt in a TTS quadruplet, the maximal abundance decrease was found from sites −2 nt (upstream) to +2 nt (downstream) flanking most TTSs (*Figure 1B*). Thus, the read ratio between −2 and +2 nts was used as the measurement of TTE, which was calculated based on "TTE = 1−[+2] / [−2]", where [−2] and [+2] represent the read abundances at −2 nt and +2 nt in Term-seq data, respectively.

Next, all identified TUs were ranked into three hierarchical groups: high TTE (> 60%), medium TTE (30% < TTE < 60%), and low TTE (< 30%) groups. Statistically, approximately 32.5%, 44%, and 23.5%

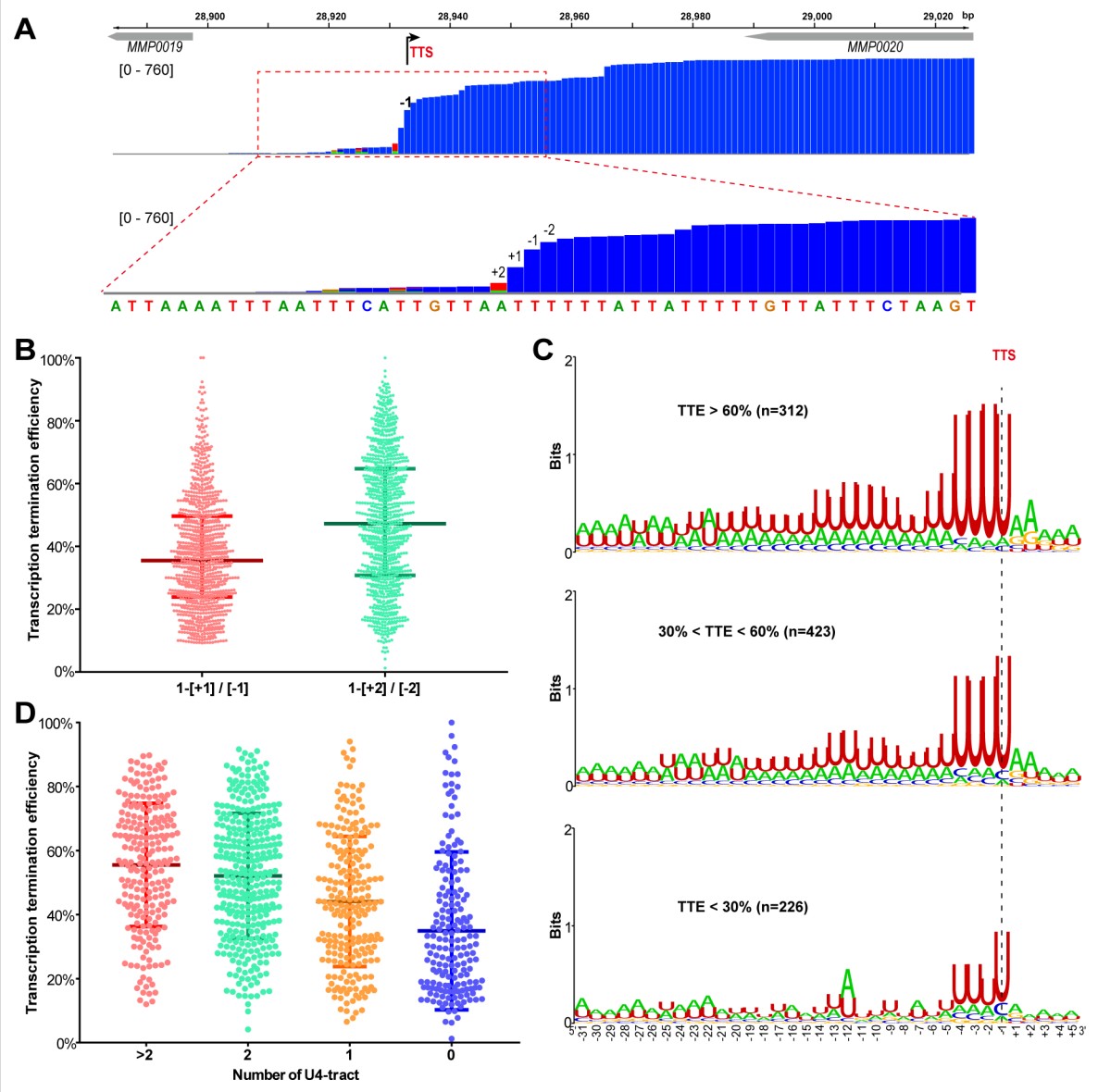

**Figure 1.** A positive correlation is observed between the terminator U4-tract numbers and the TTEs among the TUs of *M.maripaludis*. (**A**) A representative Term-seq map of *MMP0020* showing a dramatical decreasing pattern of sequencing reads at four nucleotides that flank the identified transcription termination site (TTS, -1 site indicated by bent arrow). The magnified mapping region (dotted red frame) shows reads dramatically decreasing from −2 (two nts upstream) to +2 (two nts downstream) of the TTS. The chromosome locations of the genes are indicated at the top, and the Term-seq read heights are shown in brackets. (**B**) Box-plot diagrams showing the TTE statistics of the 998 transcripts, which were calculated based on the reads ratio of nts +1 to −1 ([+1]/[−1]), and that of nts +2 to −2 ([+2]/[−2]) respectively up- and down-stream of the primary TTSs. Between the upper and lower lines are TTEs of 50% of transcripts, and the middle line represents the TTE median. (**C**) Logo representations of the terminator motif signatures in three groups of transcripts with different TTEs ( > 60%, > 30% and < 60%, < 30%). The transcript numbers of each group are indicated inside parentheses. The correlation of TTEs with the terminator U4-tract numbers was analyzed using Wilcox test, and the P values between Groups I and II, I and III, II and III were 3.4e-12, 2.22e-16 and 2.1e-5, respectively. (**D**) Box-plot diagrams showing the statistics of TTE values among the four groups of terminators that carry various numbers of U-tracts. The diagram representations are the same as those in (**B**). The statistical significance for the TTEs of the four groups analyzed by Wilcoxon rank sum test are shown in *Supplementary file 4c*.

The online version of this article includes the following source data and figure supplement(s) for figure 1:

**Source data 1.** Includes the statistic source data of *Figure 1*, *Figure 1—figure supplements 1 and 2*.

**Figure supplement 1.** Multiple TTSs in one transcript identified by Term-seq in *M. maripaludis*.

**Figure supplement 2.** Features of the primary and secondary TTSs identified in *M. maripaludis*.

**Figure supplement 3.** Term-seq maps of three representative genes, *MMP0065* (**A**), *MMP1579* (**B**), and *MMP0760* (**C**).

**Figure supplement 4.** Percentages and motifs of the TU groups that have different numbers of the terminator U4-tracts.

of TUs fell in the high, medium, and low TTE groups, respectively (*Figure 1C*). Sequence motifs, generated from −30 nt until +5 nt flanking TTSs by Weblogo, showed characteristic U-rich tracts, with each containing four consecutive uridine nucleotides (U4) preceding the TTSs among the overrepresented TUs in all the three groups (*Figure 1C*). Noticeably, a positive correlation was found between the TTE and the terminator U4-tract numbers (P<2.2e-16, spearman's cor = 0.33): two or more than two U4-tracts were found overrepresented in the high TTE group, while the U4-tract was underrepresented in the low TTE group (*Figure 1C*). To further evaluate the correlation between the U4-tracts and the TTEs, we first classified all of the defined TUs into four groups based on the U4-tract numbers preceding the primary TTSs (*Figure 1—figure supplement 4A*), and then generated the sequence motif (*Figure 1—figure supplement 4B*) and statistically calculated the TTE distribution (*Figure 1D*) in each group. Similarly, a marked positive correlation between the U4-tract numbers and the TTEs was also observed, such as TUs in the groups of >2, 2, 1, and 0 U4-tracts had the median TTEs of 55.5%, 52.1%, 43.3%, and 30%, respectively (*Figure 1D*), which demonstrated that TUs with more U4-tract numbers had higher TTEs.

These analyses suggest that the U4-tract preceding the TTS could be a key signal (motif) in dictating or affecting RNAP to pause and triggering transcription termination, and more U4-tracts could result in higher TTEs in *M. maripaludis*.

## Concurrence of the terminator U-tract and termination factor aCPSF1 in dictating high TTEs

Based on our recent finding that aCPSF1 functions as the archaeal general transcription termination factor (*Yue et al., 2020*), we quantitatively compared the Term-seq identified TTEs in the wild-type (WT) and aCPSF1 expression depleted strain (▽*aCPSF1*, a mutant retaining a residual 20% aCPSF1 abundance compared to WT at 22 °C), and found an average 50% reduction in the TTEs of primary TTSs in ▽*aCPSF1* (*Figure 2A* and *Figure 2—figure supplement 1*). Further, to quantify the contribution of aCPSF1 to TTE, the aCPSF1 dependency of a TU in transcription termination was calculated based on its TTS quadruplet read changes in ▽*aCPSF1* compared to WT using the following formula:

$$\text{TTS Quadruplet Read Ratio } (\text{TQRR}) \quad = \frac{\text{S2}\left[+2/-2\right]}{\triangledown aCPSF1\left[+2/-2\right]}.$$

That TUs having TQRR <1, that is, the read ratio between +2 and −2 nt in the TTS quadruplet is reduced due to aCPSF1 depletion, was identified as aCPSF1-dependent termination. Unexpectedly, we found that 91.6% (880/961) coding TUs have TQRR <1 (*Figure 1—figure supplement 4A* and *Supplementary file 2*), and observed an approximately linear correlation between TQRR and TTE for all studied 961 coding TUs (*Figure 2B*). This finding indicates that the majority of TUs were terminated in an aCPSF1-dependent manner, and the higher TTE of a TU, the more dependency of aCPSF1. Therefore, aCPSF1 could display a positive correlation with TTEs as well as the terminator U4-tracts and play a key role in dictating high TTEs at the genome-wide level.

Subsequently, we explored the cis-elements that may determine the aCPSF1-dependent TTEs, that is, the cis-elements recognized by aCPSF1, through statistically analyzing the correlation between the aCPSF1 dependency of TTEs and the presence of sequence motifs preceding TTSs of the coding TUs. We classified the 961 TUs into three groups based on the TQRRs: the highly aCPSF1-dependent (TQRR ≤0.6), moderately aCPSF1-dependent (0.6< TQRR < 1) and non-aCPSF1-dependent (TQRR ≥1) groups, and generated the TTS preceding sequence motif for each group using Weblogo. Interestingly, we found not only that 29.3% (282/961) and 62.2% (597/961) of TUs belonged to the highly and moderately aCPSF1-dependent groups respectively, and only 8.4% (81/961) of TUs belonged to the aCPSF1-independent group, but also a significant positive correlation between the aCPSF1-dependency and the numbers of U4-tract preceding TTSs, namely, the higher aCPSF1 dependency, the more featured U4-tracts of the TU groups (*Figure 2C*). Additionally, we evaluated the relationship between the aCPSF1-dependency and the above four U4-tract TU groups analyzed in *Figure 1D*. We found that 94.5% (736/779) of TUs in the TU groups with ≥1 U4-tracts have a TQRR <1 (*Figure 1—figure supplement 4A*), and TU groups with ≥2, 2, 1, and 0 U4-tracts had median TQRRs of 0.655, 0.67, 0.76, and 0.92, respectively. These findings indicate that the majority of TUs with U4-tract depend on aCPSF1 for termination and the TU groups containing more U4-tracts have lower median TQRR values, namely, higher aCPSF1-dependency (*Figure 2D*). Noteworthily, even 79.1% (144/182) of TUs

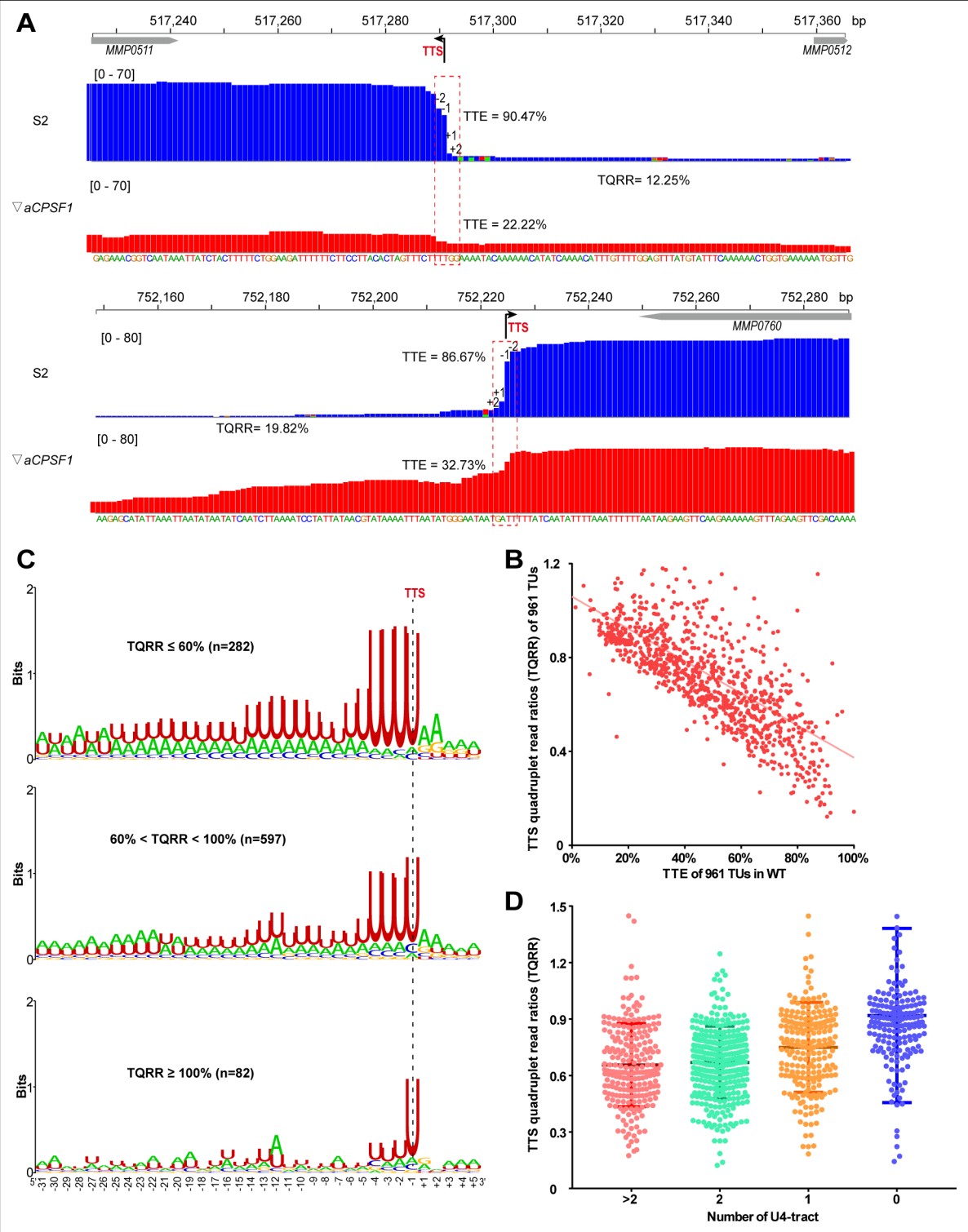

**Figure 2.** Co-occurrence of aCPSF1 and the terminator U4-tract is correlated with the genome-wide TTEs of *M. maripaludis*. (**A**) Visualized Term-seq read maps of the representative genes, *MMP0511* (top) and *MMP0760* (bottom), show sharper reads decreasing between the −2 and +2 nts (dotted frame) respectively down- and up-stream of the TTSs (-1) in the wild-type (WT) strain (S2) than in the aCPSF1 depletion mutant (▽aCPSF1). The chromosome locations of the genes are indicated at the top. The bent arrow indicates the Term-seq identified TTS. TQRR represents the TTS quadruplet read ratio of a TU in WT (S2) to that in ▽aCPSF1, with the lower values representing a higher aCPSF1 dependency of a TU in transcription termination. The mapping read heights are shown inside the brackets. The TTE is calculated as above. (**B**) A linear correlation is observed between the TQRRs and TTEs of 961 protein coding TUs. (**C**) Logo representations of the terminator motif signatures are shown for highly aCPSF1-dependent

*Figure 2 continued on next page*

*Figure 2 continued*

(TQRR ≤60%), moderately aCPSF1-dependent (60%< TQRR < 100%) and non-dependent (TQRR ≥100%) groups. The TU numbers of each group are shown in parentheses. (**D**) Box-plot diagrams showing the TQRR (aCPSF1 dependency) statistics of the terminators carrying >2, 2, 1, and 0 U4-tracts. Between the upper and lower lines are TQRRs of 50% of transcripts, and the middle line represents the TQRR median. The statistical significance for the TQRRs of the four groups analyzed by Wilcoxon rank sum test are shown in ***Supplementary file 4d***.

The online version of this article includes the following source data and figure supplement(s) for figure 2:

**Source data 1.** Includes the statistic source data of ***Figure 2***, ***Figure 2—figure supplements 1 and 2***.

**Figure supplement 1.** Box plot diagram showing the statistics of TTEs of the Term-seq identified primary TTSs in the WT (S2) and ▽*aCPSF1* mutant.

**Figure supplement 2.** The terminator motif and the linear correlation of TTEs and TQRRs of noncoding RNAs in *M. maripaludis*.

**Figure supplement 3.** Transcription readthrough (TRT) of noncoding RNAs caused by the depletion of *aCPSF1*.

**Figure supplement 3—source data 1.** Includes the bolt source date of ***Figure 2—figure supplement 3***.

with 0 U4-tract had a TQRR <1 (***Figure 1—figure supplement 4A***), suggesting that the transcription of these TUs with weak terminator can also be terminated under the assistance of aCPSF1. Additionally, among the 8.4% (81/961) of TUs that fell into the aCPSF1-independent group, only 3.95% (38 of 961) had 0 U4-tract (***Figure 1—figure supplement 4A***), suggesting that these very few TUs may be terminated by mechanisms independent of both aCPSF1 and U4-tract terminator, or that the TTSs identified for these TUs are potential RNA processing sites derived from stale RNA processing or degradation products.

Similar transcriptional termination features were also found in the non-coding RNAs as follows: (i) a shorter U-tract terminator motif (U-tract) preceding the TTSs (***Figure 2—figure supplement 2A*** and ***Supplementary file 3***), (ii) a positive correlation between the TTE and the terminator U-tract length (***Figure 2—figure supplement 2B***), and (iii) a linear correlation between TQRRs and TTSs (***Figure 2—figure supplement 2C***). Consistently, through querying the transcription pattern of two representative noncoding RNAs, we found prolonged transcript 3'-ends (***Figure 2—figure supplement 3A***) and detected transcription readthrough using Northern blot (***Figure 2—figure supplement 3B***). These findings indicate that transcription termination of non-coding RNAs could resemble that of coding TUs and depend on both the U-tract terminator and termination factor aCPSF1 as well.

Collectively, these results indicate that both the terminator cis-element U4-tracts and the trans-action termination factor aCPSF1 exhibit high positive correlation with the TTEs, and the two appear to cooperatively dictate high TTEs at a genome-wide level in vivo.

## aCPSF1 specifically binds to RNAs embedding the terminator U4-tract sequence in vitro

The collaboration of aCPSF1 and the terminator U-tract in dictating TTE suggests that this termination factor may specifically recognize the terminator U-tract motif embedded in the nascent transcript 3'-end to dictate archaeal transcription termination. To test this hypothesis, we first assayed the binding ability of aCPSF1 to three synthetic RNAs that contain the terminator U-tract sequences of the transcripts *MMP0901*, *MMP1149*, and *MMP1100*, which were determined to be cleaved by the recombinant aCPSF1 in our previous study (***Yue et al., 2020***). An RNA fragment with the transcript 3'-end sequence of *MMP1697* lacking a U-tract was included as a control. Using RNA electrophoretic mobility shift assay (rEMSA), shifted protein–RNA complex bands could be observed in the three U-tract containing RNAs, but not in that without U-tract at the same concentrations of aCPSF1 (***Figure 3—figure supplement 1***). Next, 12 additional RNA fragments in a consensus 36 nt length and of transcript 3'-end sequences of genes listed in ***Figure 3***, from 30 nts upstream to 5 nts downstream of the TTS, were used to compare the binding ability of aCPSF1. These sequences were derived from six transcripts embedding ≥2 U4-tracts (***Figure 3A***), three transcripts embedding 1 U4-tract (***Figure 3B***) and three with no (0) U4-tract (***Figure 3C***), respectively. The rEMSA results indicated that aCPSF1 exhibited the strongest binding to those with ≥2 U4-tracts, an average ~5 fold weaker binding to those with 1 U4-tract and weakest binding to those with 0 U-tract (***Figure 3*** and ***Figure 3—figure supplement 2***). Supportively, through surface plasmon resonance (SPR) assay using the same concentrations of aCPSF1, the highest and lowest resonance units (RUs) were determined for the RNA containing the longest U-tract from the *MMP0400* 3'-end and that carrying the shortest U-tract from *MMP1406* 3'-end, respectively (***Figure 3—figure supplement 3***). Therefore, these results

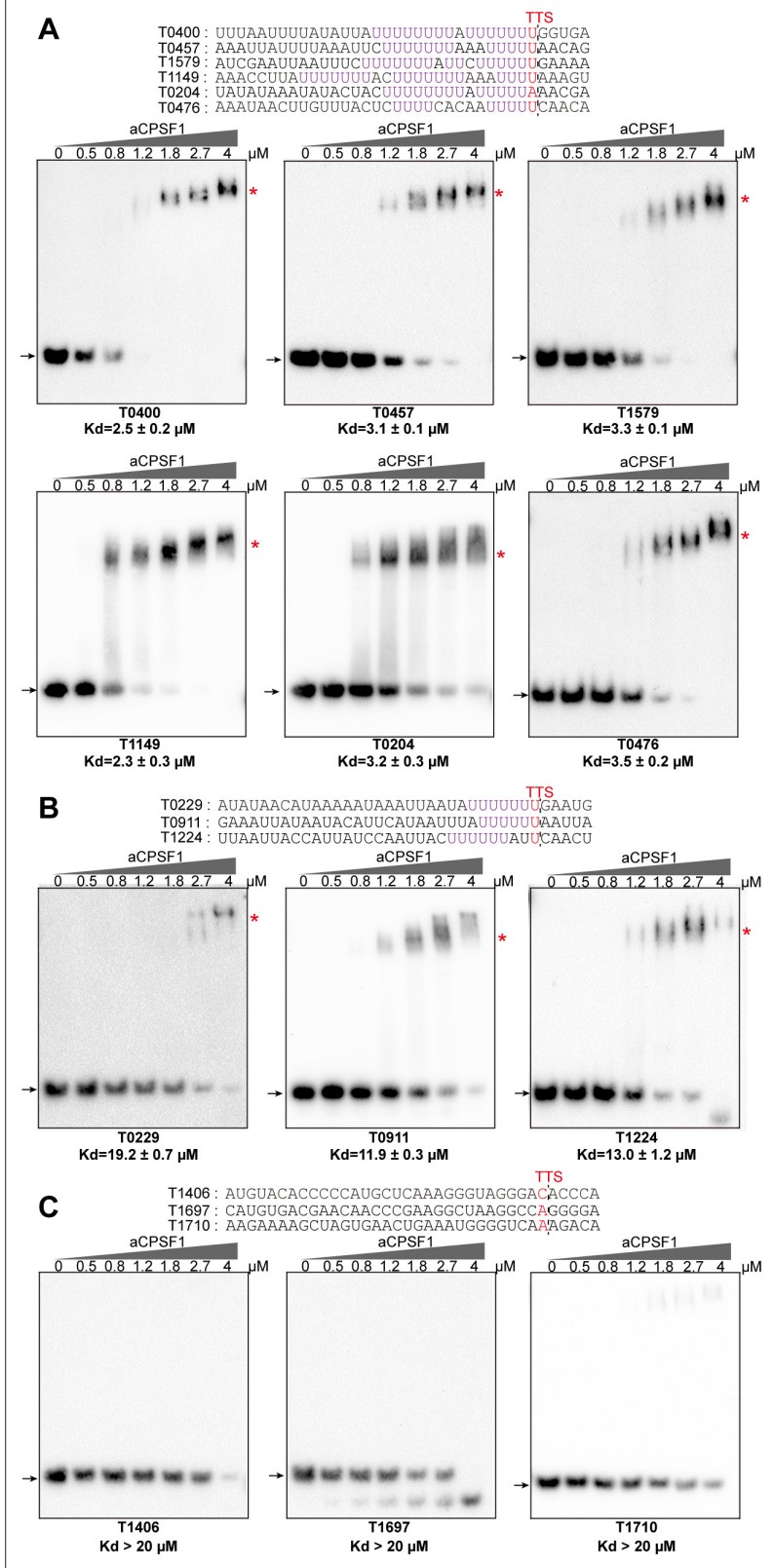

**Figure 3.** Binding specificity of aCPSF1 to RNAs carrying different numbers of U4-tracts determined by rEMSA assays. RNAs with a consensus length of 36 nt derived from the indicated gene terminators that carry ≥2 U4-tracts (**A**), 1 U4-tract (**B**), and 0 U4-tract (**C**) were used as the binding substrates. RNA sequences are shown in the top panels with red letters indicating Term-seq identified TTSs. The gradient concentrations of aCPSF1 used in the

*Figure 3 continued on next page*

*Figure 3 continued*

binding reactions are indicated at the top of gels. Detailed binding procedure is described in the Materials and methods section. The arrows and red asterisks indicate the free RNA substrates and the shifted RNA-aCPSF1 complexes, respectively. The binding assay for each RNA substrate was performed in triplicate. Equilibrium dissociation constants (Kd) were calculated from the binding curves based on the quantification of unbound and bound substrates, and the average Kd and standard deviations are shown.

The online version of this article includes the following source data and figure supplement(s) for figure 3:

**Source data 1.** Includes the gel source data of *Figure 3*.

**Figure supplement 1.** The binding specificity of aCPSF1 to U-tract RNAs determined by rEMSA assays.

**Figure supplement 1—source data 1.** Includes the gel source data of *Figure 3—figure supplement 1*.

**Figure supplement 2.** Binding curves of aCPSF1 to indicated RNAs.

**Figure supplement 3.** The binding specificity of aCPSF1 to U-tract RNAs assayed by SPR.

**Figure supplement 3—source data 1.** Includes the statistic source data of *Figure 3—figure supplement 3*.

---

demonstrated that aCPSF1 specifically recognizes the transcripts with U-tract sequences and binds preferentially to the transcripts carrying more U4-tracts at the 3'-ends.

Next, the minimal RNA length and U-tract sequence required for aCPSF1 binding were investigated. The 36 nt RNA sequences embedding ≥2 U4-tracts from the transcript 3'-ends of *MMP0204* and *MMP0400*, were sequentially truncated by six nts from the 5'-end to generate 30 nt, 24 nt, and 18 nt RNA substrates. The rEMSA results indicated that aCPSF1 bound to the 36 nt, 30 nt and 24 nt RNAs with similar affinity, but had an average ~2 fold reduced affinity to the RNA of 18 nt (*Figure 4A* and *Figure 4—figure supplements 1A and 2*). To confirm the role of the U-tracts in determining the binding specificity of aCPSF1, base mutation was performed on either one U-tract (18nt-M1 and 18nt-M2) or both U-tracts (18nt-M3) in the 18 nt RNAs. The rEMSA results indicated that mutation of either one U4-tract or both U-tracts remarkably reduced the binding ability of aCPSF1 to the RNA substrate (*Figure 4B* and *Figure 4—figure supplements 1B and 2*). Reciprocally, by mutating two As to Us to increase 1 to 2 U-tracts on the RNA with the *MMP0229* 3'-end sequence (T0229-18nt and T0229-18nt-M1, respectively), the binding affinity of aCPSF1 was notably increased compared to T0229-18nt, whereas mutation of the four Us to four As at T0229-18 nt to obtain an RNA sequence lacking a U-tract (T0229-18nt-M2) completely abolished aCPSF1 binding (*Figure 4C*). Furthermore, the footprint assay was performed on the above used RNA with the T0204 terminator sequence using RNase I digestion in the absence or presence of aCPSF1, and a clear footprint of aCPSF1 on the U-tract region was found (*Figure 4—figure supplement 3*). These results demonstrated that both of the two U-tracts in the transcript 3'-end are necessary for efficient and specific binding of aCPSF1, and that the U-tract region is the exact region that aCPSF1 binds to. Following, the minimum length of the consecutive uridines in the two U-tracts was evaluated. U to C mutation was performed on the two U-tracts RNA of T0204-24nt to shorten the length of the consecutive uridines to generate T0204-DU5, T0204-DU4, and T0204-DU3, which carry 5, 4, or 3 consecutive uridines in each U-tract, respectively. The rEMSA results indicated that aCPSF1 efficiently bound to T0204-DU5 and T0204-DU4, but could not bind to T0204-DU3. This suggests that the two U4-tracts is the minimum terminator sequence for efficient binding of aCPSF1 (*Figure 4D*). Collectively, the in vitro RNA binding experiments demonstrated that an RNA fragment embedding a two U4-tracts and with a minimum length of 18 nt is the cis-element required by the termination factor aCPSF1 for efficient and specific binding.

## In vivo cooperation of aCPSF1 and the terminator U-tract dictates the effective transcription termination

To verify the in vivo cooperative action of aCPSF1 and the terminator U-tract in dictating TTEs, indispensability of either one for efficient transcription termination was assayed. A dual reporter gene system was constructed as shown in *Figure 5A*. Five terminators in a length of 36 nt and carrying the same sequences used in the rEMSA assays (*Figure 3*), were each inserted between the genes encoding luciferase and mCherry fluorescent proteins, those are T1149 and T0204 carrying 2, T0229 and T0911 carrying 1, and T1710 carryings 0 U-tracts. The reporter constructs were respectively transformed into the WT strain S2, ▽*aCPSF1*, and two complementary strains, Com(WT) and Com(Mu) that each carry the wild-type aCPSF1 and its catalytically inactive mutant complementation in ▽*aCPSF1*,

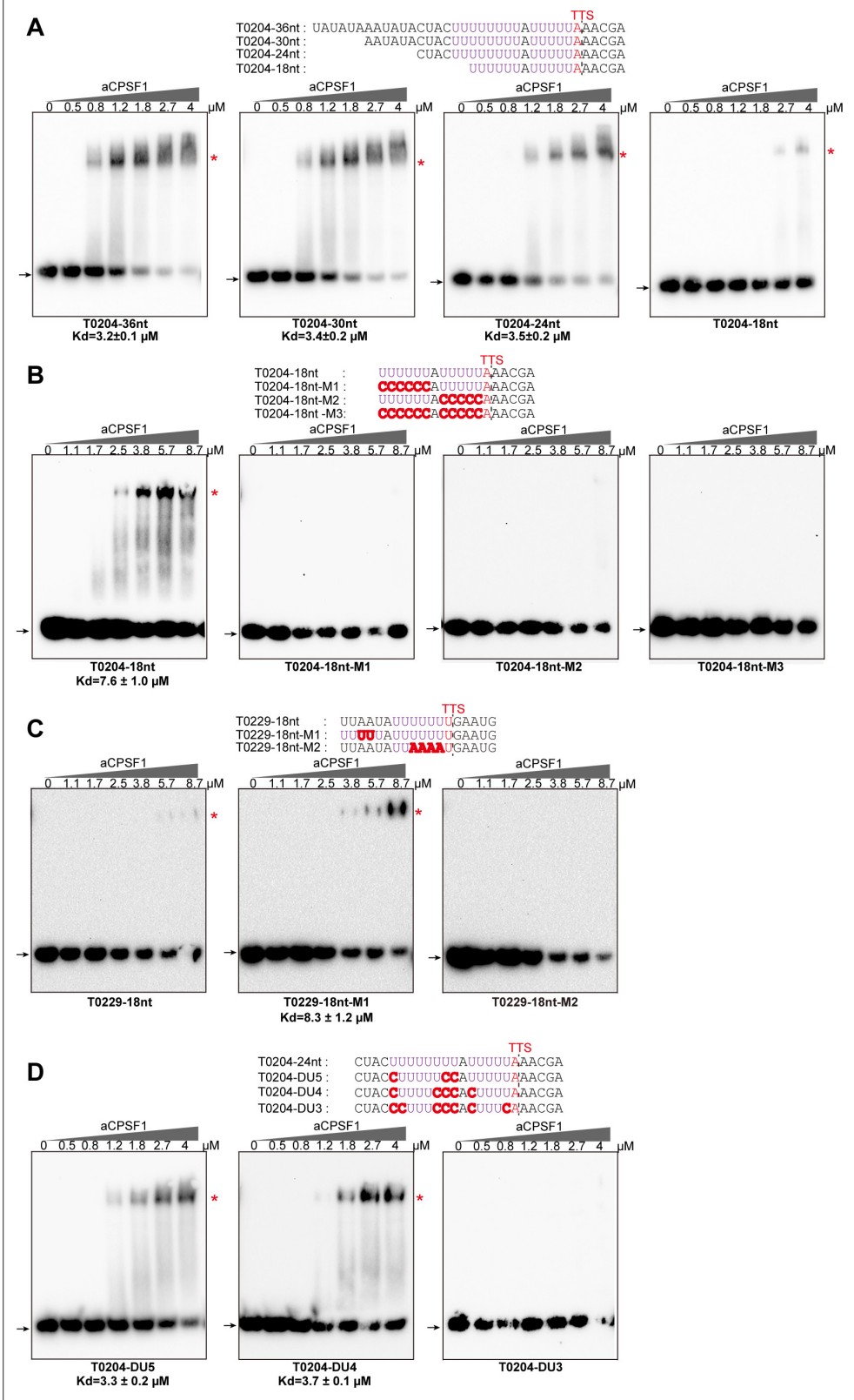

**Figure 4.** The minimal RNA length and U-tract base stringency required for the specific binding of aCPSF1 determined by rEMSA assays. RNAs with indicated lengths and base mutations shown in the top panels that are derived from the native terminator sequences of *MMP0204* (T0204) and *MMP0229* (T0229) were used as the binding substrates of aCPSF1. (**A**) The T0204 RNA with a length of 36 nt was truncated by six nts from the 5′ end,

*Figure 4 continued on next page*

*Figure 4 continued*

resulted in RNAs with lengths of 30 nt, 24 nt, and 18 nt. (**B**) The 18nt-T0204 RNA was base mutated in either one (**M1 and M2**) or both U-tracts (**M3**). (**C**) A one U-tract RNA derived from the T0209 was base mutated to construct a two U-tract mutant (**M1**) and a non-U-tract mutant (**M2**). (**D**) The T0204 RNA that contains two U-tracts was base mutated to generate the two U-tracts with each U-tract containing 5 (DU5), 4 (DU4), and 3 (DU3) consecutive Us, respectively. TTSs identified by Term-seq and the mutated residues are shown as plain and bold red letters, respectively. The gradient concentrations of aCPSF1 used in the binding reactions are indicated at the top of the gels. The detailed rEMSA procedure is described in the Materials and methods section. The arrows and red asterisks indicate the free RNA substrates and the RNA-aCPSF1 complexes, respectively. The binding assay for each RNA substrate was performed in triplicate. Equilibrium dissociation constants (Kd) were calculated from the binding curves based on the quantification of unbound and bound substrates, and the average Kd and standard deviations are shown.

The online version of this article includes the following source data and figure supplement(s) for figure 4:

**Source data 1.** Includes the gel source data of *Figure 4*.

**Figure supplement 1.** The minimal RNA length and U-tract base stringency required for the specific binding of aCPSF1 on T0400 terminator determined by rEMSA assays.

**Figure supplement 1—source data 1.** Includes the gel source data of *Figure 4—figure supplement 1*.

**Figure supplement 2.** Binding curves of aCPSF1 to indicated RNAs.

**Figure supplement 3.** The RNase footprint assay identifies the binding region of aCPSF1 to the T0204 RNA carrying two terminator U-tracts.

**Figure supplement 3—source data 1.** Includes the gel source data of *Figure 4—figure supplement 3*.

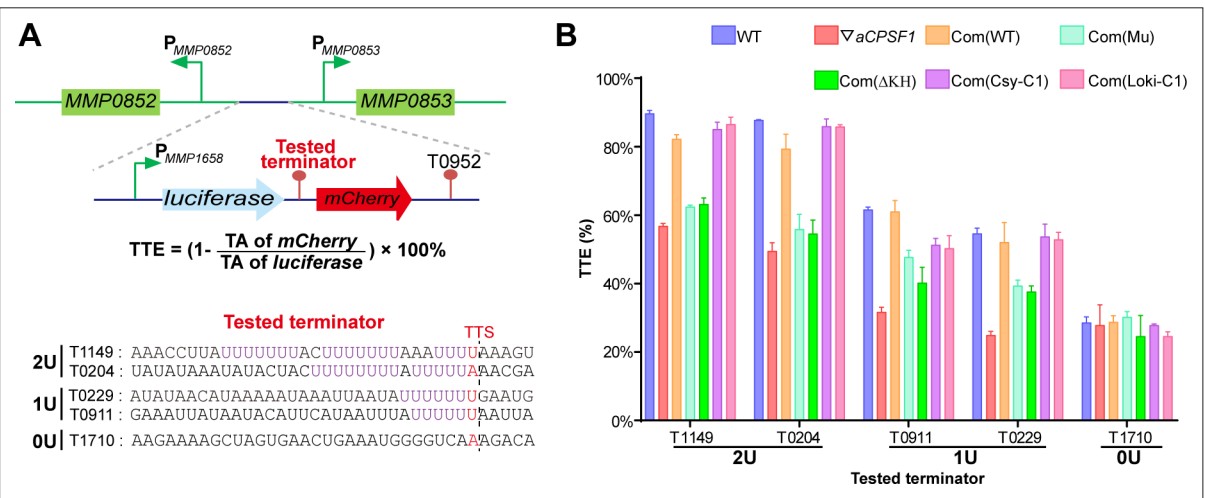

**Figure 5.** Terminator reporter system demonstrates TTE variations in co-occurrence and either absence of aCPSF1 and the terminator U-tract. (**A**) Schematic depicting the construction of the terminator reporter system. The tested terminator sequences carrying different numbers of U-tracts were each inserted between the upstream luciferase and downstream mCherry genes, and then fused downstream of the *MMP1658* promoter (P_MMP1658) and upstream of the *MMP0952* terminator (T0952). Subsequently, the constructed DNA fragment was inserted in between the promoters of *MMP0852* and *MMP0853* in the genomes of various strains listed in (**B**) through homologous recombination. The detailed protocol is described in the Materials and methods section. The formula for TTE calculation and the sequences of the tested terminators are shown below. (**B**) Quantitative RT-PCR was conducted to quantify the abundance of the luciferase and mCherry transcripts in various terminator constructs, which were expressed in the wild-type strain (WT), the aCPSF1 depletion mutant (∇aCPSF1), and the ∇aCPSF1 strain complemented with wild-type (Com(WT)), KH domain truncated (Com (ΔKH)), and catalytic mutated (Com(Mu)) mutants of the *M. maripaludis* aCPSF1 (*Mmp*-aCPSF1), and the aCPSF1 orthologs from Ca. *Lokiarchaeum* sp. GC14_75 Com(*Loki*-aCPSF1) and Ca. *Cenarchaeum symbiosum* Com(*Csy*-aCPSF1). The TTEs were calculated based on the formula in the middle panel of (**A**), and TA indicates transcription abundance. Triplicated cultures were assayed and the averages and standard derivations are shown. 2 U, 1 U, and 0 U indicate the tested terminators carrying 2 (T1149, T0204), 1 (T0911, T0229), and 0 (T1710) U4-tracts, respectively. The statistical significance of the qRT-PCR data in different genetic strains vs. WT were determined by T-test and are shown in *Supplementary file 4e*.

The online version of this article includes the following source data for figure 5:

**Source data 1.** Includes the statistic source data of *Figure 5*.

respectively. The TTE was calculated based on the transcript abundances (TAs) of the downstream mCherry and the upstream luciferase using the formula: $\text{TTE} = 1 - \frac{\text{TA(mCherry)}}{\text{TA(luciferase)}}$. TA was estimated by quantitative reverse-transcription (RT-qPCR), as the fluctuated fluorescence intensity precluded it to be measured. It determined that the TTEs were highest for the terminators with 2 U4-tracts (T1149 and T0204), lower for those with 1 U4-tract (T0911 and T0229), and the lowest for that with 0 U-tract (T1710) in WT (*Figure 5B*). Therefore, the reporter system confirms the positive correlation between the numbers of terminator U-tract and TTEs in vivo.

However, in ▽*aCPSF1*, terminators containing either 2 U4- (T1149 and T0204) or 1 U4-tracts (T0911 and T0229) all exhibited reduced TTEs, while the terminator with 0 U-tract (T1710) retained nearly the same TTE as that in the WT strain (*Figure 5B*). We then calculated the aCPSF1 dependency in the reporter assays by the formula of $\text{TA Ratio} = \frac{\text{S2}\left[\text{TA(mCherry)}/\text{TA(luciferase)}\right]}{\triangledown aCPSF1\left[\text{TA(mCherry)}/\text{TA(luciferase)}\right]}$, which is very similar to that used for the TQRR calculation. TA ratios of 0.24, 0.24, 0.56, 0.61, and 0.99 were determined for the terminators carrying 2 U4-tracts (T1149 and T0204), 1 U4-tract (T0911 and T0229), and 0 U-tract (T1710), respectively. This further verifies that terminators carrying more U4-tracts have a higher aCPSF1-dependency. It is worth noting that similar reduced TTEs were determined for the U-tract containing terminators in the Com(Mu) strain as those in ▽*aCPSF1*, while the TTEs in the Com(WT) strain were similar to those in the WT (*Figure 5B*). This indicates that the nuclease activity of aCPSF1 is also essential to the TTE dictation of the U-tract containing terminators. Thus, the dual gene reporter assays verified the in vivo cooperation of the termination factor aCPSF1 with the terminator U-tract in dictating effective TTE in *M. maripaludis*, and demonstrated that the nuclease activity of aCSPF1 is necessary for the TTE dictation.

## The N-terminal KH domains equip aCPSF1 to specifically bind the terminator U-tract and dictate TTE

The resolved crystal structure of an aCPSF1 ortholog from *Methanothermobacter thermautotrophicus* revealed a tripartite architecture consisting of two N-terminal KH domains (KHa and KHb), a central MβL domain, and a C-terminal β-CASP domain (*Silva et al., 2011*). KH domains have been predicted to be involved in RNA sequence recognition and binding (*Phung et al., 2013*; *Silva et al., 2011*). To assess whether the KH domains equip aCPSF1 to recognize the terminator U-tract, the N-terminal fragment of *Mmp*-aCPSF1, with a length of 149 amino acids and containing the two KH domains, was expressed. The purified KH protein was first assayed for the RNA binding abilities. Resemble to the intact aCPSF1 protein, the recombinant KH domains exhibited higher binding affinity to the RNAs containing longer or more U-tracts (T0204, T0400, T0457, T1579, and T0229) than to those with no U-tract (T1406 and T1697) (*Figure 6A*). Therefore, the N-terminal KH domains enable aCPSF1 to specifically recognize the consecutive U-tracts embedded in the transcript 3'-ends.

To further evaluate the role of the aCPSF1 KH domains in the in vivo transcription termination, we complemented into ▽*aCPSF1* with the *aCPSF1* gene lacking the KH domains to obtain the complementary strain Com(ΔKH). Although the truncated ΔKH-aCPSF1 and an even ~1.5-fold increase of the whole-length aCPSF1 were detected in Com(ΔKH) (*Figure 6B*), the same growth defect as ▽*aCPSF1* was found for Com(ΔKH), in contrast to the strain Com(WT), which carries the wild-type aCPSF1 complementation and restored the growth defect of ▽*aCPSF1* (*Figure 6C*). Furthermore, 3'RACE assays detected similar transcription readthrough (TRT, transcription termination defect) products in seven transcripts (*MMP0901*, *MMP1149*, *MMP0400*, *MMP0457*, *MMP1579*, *MMP0229*, and *MMP1224*) with ≥1 U-tract embedded at the 3'-ends in Com(ΔKH) as that in ▽*aCPSF1*, while no TRT products were detected in these seven transcripts in Com(WT) (*Figure 6D* and *Figure 6— figure supplement 1*). In contrast, no TRTs were found in the transcripts (*MMP1710* and *MMP1406*) without 3'-end U-tracts, in either Com(ΔKH) or ▽*aCPSF1* (*Figure 6D*). Moreover, using similar assays to those described in *Yue et al., 2020*, the gel filtration coupled with western blot showed that loss of KH domain did not impair the interaction of aCPSF1 with RNAP (*Figure 6—figure supplement 2*). Together, these results indicated that in contrast to the complemented wild-type aCPSF1, although the complemented ΔKH-aCPSF1 still interacts with RNAP, it cannot restore the growth and transcription termination defect of ▽*aCPSF1*. These finding suggest that the KH domains are essential for aCPSF1 to perform regular transcription termination of the transcripts carrying U-tract terminators and maintain the normal growth of *M. maripaludis*.

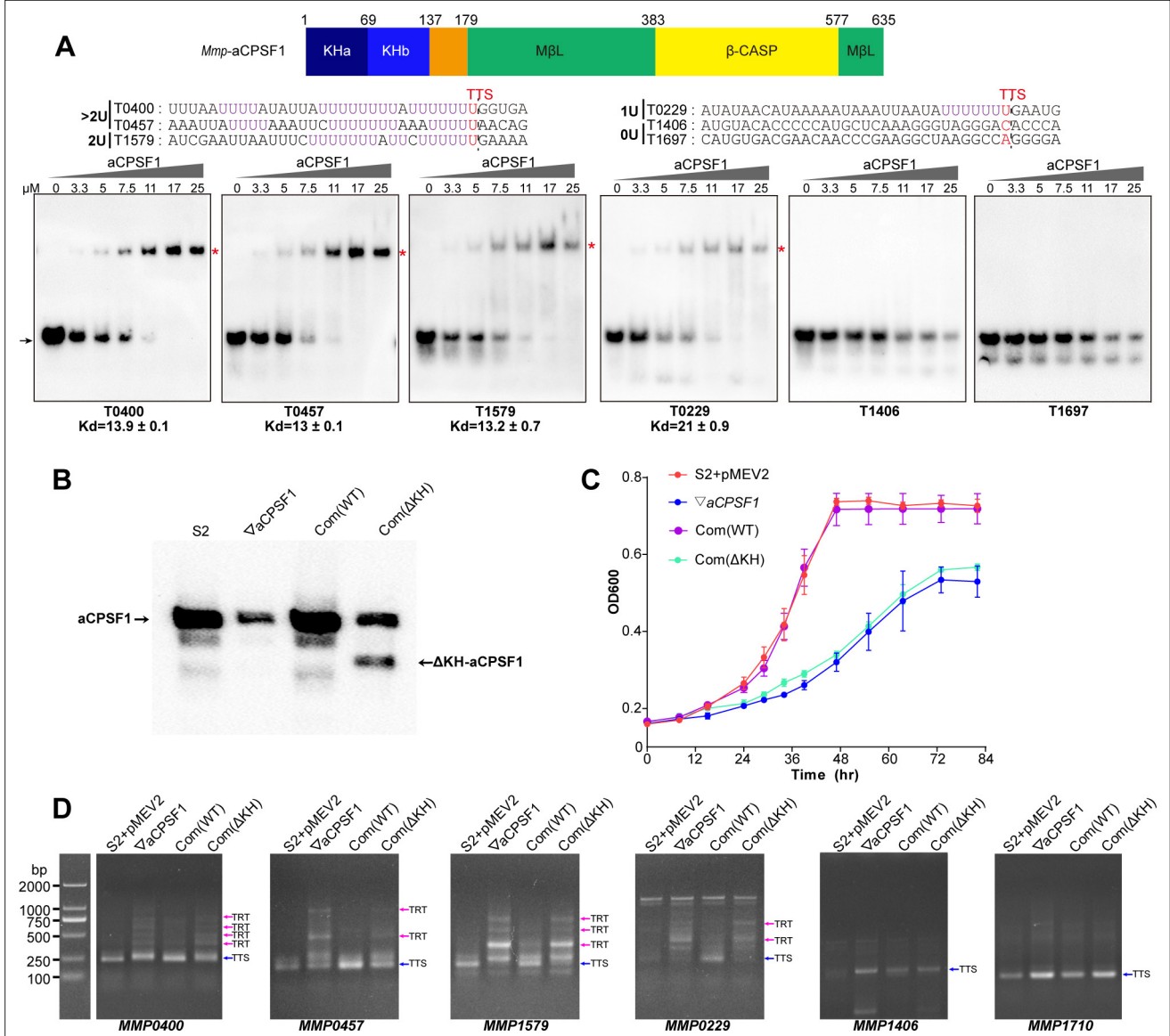

**Figure 6.** The N-terminal KH domains of aCPSF1 contribute to the binding specificity to terminator U-tracts and transcription termination. (**A**) Schematic (upper panel) showing the aCPSF1 protein architecture with two N-terminal KH domains, the central MβL domain and the C-terminal β-CASP domain. rEMSA assay (lower panel) was performed to determine the binding specificity of the KH domains to the terminators carrying different numbers of U4-tract as indicated. >2 U, 2 U, 1 U, and 0 U indicate the tested RNAs carried >2, 2, 1, and 0 U4-tracts, respectively. The assayed RNA sequences are shown at the top of the gels with Term-seq identified TTSs shown in red. rEMSA was performed as in *Figure 3*, and the gradient concentrations of aCPSF1 KH domain are indicated. The arrows indicate free RNAs and red asterisks indicate the RNA–KH complexes. (**B**) Western blot demonstrates the expression of the intact and KH domain deleted aCPSF1 (aCPSF1-ΔKH) proteins in the wild-type strain S2 that carries the empty complementation plasmid of pMEV2 (S2+ pMEV2), the aCPSF1 depletion mutant (∇aCPSF1), the ∇aCPSF1 complemented with the wild-type (Com(WT)), and KH domain truncated *Mmp*-aCPSF1 (Com(ΔKH)). The arrows indicate the respective proteins. (**C**) Growths of the four strains were assayed on three batches of 22°C-grown cultures, and the averages and standard deviations are shown. (**D**) 3'RACE assays detected the transcriptional readthroughs (TRTs) in 22°C-grown strains using the same symbols as in (**B**). Blue and magenta arrows indicate the PCR products of normal terminations (TTSs) and transcriptional readthroughs (TRTs), respectively. M, a DNA ladder is shown on the left (lane M).

The online version of this article includes the following source data and figure supplement(s) for figure 6:

**Source data 1.** Includes the gel and bolt source data of *Figure 6*.

**Source data 2.** Includes the statistic source data of *Figure 6C*.

**Figure supplement 1.** 3'RACE assays detected transcriptional readthroughs (TRTs) of *MMP1149*, *MMP0901* and *MMP1224* in 22 °C -grown wild-type strain (S2), *aCPSF1* depletion mutant (∇aCPSF1), and the ∇aCPSF1 complementated with the wild-type (Com(WT)) and KH domain truncated aCPSF1 (Com(ΔKH)) respectively.

*Figure 6 continued on next page*

*Figure 6 continued*

**Figure supplement 1—source data 1.** Includes the gel source data of *Figure 6—figure supplement 1*.

**Figure supplement 2.** Interaction of the KH domain deleted aCPSF1 (ΔKH-aCPSF1) with RNA polymerase detected by gel filtration coupled with western blot.

**Figure supplement 2—source data 1.** Includes the bolt source data of *Figure 6—figure supplement 2*.

**Figure supplement 3.** Assays that determines the vital role of the dimerization of aCPSF1 in transcription termination.

**Figure supplement 3—source data 1.** Includes the bolt and gel source data of *Figure 6—figure supplement 3A and C*.

**Figure supplement 3—source data 2.** Includes the statistic source data of *Figure 6—figure supplement 3B*.

To further evaluate the contributions of the KH domains to aCPSF1-dependent TTEs in the transcripts containing terminator U-tracts in vivo, the five terminator reporters constructed above were transformed into Com(ΔKH). The TTEs of these reporters in Com(ΔKH) were determined as described above, and the result showed that in contrast to the strain Com(WT), complementation with the truncated ΔKH-aCPSF1 could not restore the TTE reductions in all of the tested transcripts containing U-tract terminators (*Figure 5B*, Com(ΔKH)). Therefore, both the in vitro and in vivo experiments determined that the N-terminal KH domains are necessary for aCPSF1 to specifically recognize and collaborate with the terminator U-tracts in dictating effective TTEs.

## Dimerization is vital for aCPSF1 to implement transcription termination

The crystal structure study indicated that the aCPSF1 orthologs present as dimer in solution (*Mir-Montazeri et al., 2011*; *Silva et al., 2011*), and the dimerization was determined to be functionally important in the in vitro RNA binding, and exonucleolytic and endonucleolytic activities by deleting the 12 conserved residues at C-terminal to disrupt the dimerization in *Pyrococcus abyssi* aCPSF1 (*Phung et al., 2013*). Therefore, to determine whether the dimerization is important for aCPSF1 in transcription termination, we tested the complementary function of the ΔC13 variant of *mmp*-aCPSF1 in ▽aCPSF1 by constructing the strain Com(ΔC13). Although western blot displayed successful expression of the complemented *mmp*-aCPSF1-ΔC13 variant (*Figure 6D* and *Figure 6—figure supplement 3A*), in contrast to the strain Com(WT), complementation of *mmp*-aCPSF1-ΔC13 could not restore the growth and transcription termination defects of ▽aCPSF1 (*Figure 6D* and *Figure 6—figure supplement 3B and C*). Therefore, these results indicate that the dimerization of aCPSF1 is vital for it to function in transcription termination.

## The strategy of aCPSF1 cooperated with the terminator U-tract in dictating TTE may be widely employed in archaea

In archaea, the U-rich terminators with 5, 6, and 7 consecutive Us (U5, U6, and U7) have been reported to be widely distributed in transcript 3′-ends (*Berkemer et al., 2020*; *Dar et al., 2016a*; *Maier and Marchfelder, 2019*; *Yue et al., 2020*); meanwhile, the aCPSF1 orthologs have also been found to be strictly conserved and appear being vertically inherited (*Li et al., 2021*; *Phung et al., 2013*; *Yue et al., 2020*). Therefore, we questioned whether the cooperative mode of aCPSF1 and the terminator U-tract is a common termination mechanism in archaea. To test this hypothesis, two aCPSF1 orthologs, *Loki*-aCPSF1, from Ca. *Lokiarchaeum* sp. GC14_75 belonging to Lokiarchaeota, and *Csy*-aCPSF1, from Ca. *Cenarchaeum symbiosum* affiliated with Thaumarchaeota, were chosen. Both were determined to be capable of implementing the transcription termination function in *M. maripaludis*, although they exhibit only 48% and 40% amino acid sequence identities with *Mmp*-aCPSF1, respectively (*Yue et al., 2020*). The above five tested terminator reporters were transformed into strains Com(*Loki*-aCPSF1) and Com (*Csy*-aCPSF1) that were generated by complementing the *Loki*-aCPSF1 and *Csy*-aCPSF1 into ▽aCPSF1, respectively (*Yue et al., 2020*). Through quantifying transcript abundances of the reporter genes, the TTEs of the tested terminators in Com(*Loki*-aCPSF1) and Com(*Csy*-aCPSF1) were calculated as described above. Noticeably, in strains Com(*Loki*-aCPSF1) and Com(*Csy*-aCPSF1), a similar U4-tract number related TTE pattern was determined as that in strain Com(WT) that contains *Mmp*-aCPSF1 (*Figure 5B*, Com(Loki-C1), Com(Csy-C1), and Com(WT)), that is, the highest TTEs were determined for terminators with 2 U4-tracts (T1149 and T0204), and the TTEs with 1 U4-tract (T0911 and T0229) were lower. This demonstrated that the aCPSF1 orthologs, although from distantly related archaea, could dictate the TTEs in *M. maripaludis* via distinguishing the terminator U-tracts. Given

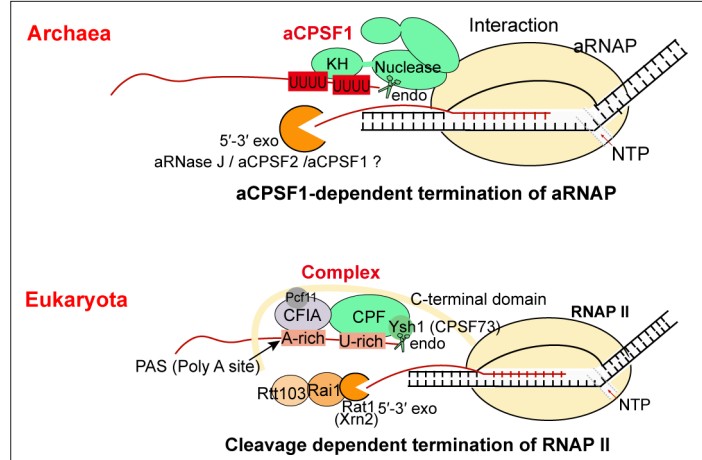

**Figure 7.** The aCPSF1-dependent archaeal transcription termination represents the archetype of the eukaryotic RNAP II termination mode. The general archaeal transcription termination factor aCPSF1, relying on the N-terminal KH domains specifically recognizing the terminator U-tract and the nuclease domain cleaving at the 3'-end, triggers transcription termination of archaeal RNA polymerase (aRNAP) (upper panel). The representative budding yeast RNAP II transcription termination model (*Kuehner et al., 2011*; *Porrua et al., 2016*; *Porrua and Libri, 2015*) is shown in the lower panel.

the ubiquitous distributions of both the termination factor aCPSF1 and the cis-element terminator U-tracts, the cooperative action mode of the two in dictating TTEs is predicted to be widely employed in various archaeal phyla.

## Discussion

Although transcription termination is a fundamental biological process in controlling transcript boundaries (*Porrua et al., 2016*; *Ray-Soni et al., 2016*), understanding of archaeal transcription termination mechanisms is limited to the 3'-end U-tract cis-element and the recently reported aCPSF1 (FttA), a protein that controls the genome-wide transcription termination and cellular fitness (*Sanders et al., 2020*; *Yue et al., 2020*). While, aCPSF1-dependent termination was assumed as just a back-up mechanism for genes/operons that contain weak intrinsic termination signals (*Sanders et al., 2020*; *Wenck and Santangelo, 2020*). In this study, through systematic analysis of the transcript 3'-end sequencing data in a genetically tractable archaeon *M. maripaludis*, we found that aCPSF1 cooperatively with the terminator U-tracts dictates the genome-wide in vivo TTEs, but not restricted to the TUs with weak intrinsic termination signals. Further, this work has elucidated that aCPSF1, functioning as a dimer and through its N-terminal KH domains, specifically binds to the terminator U-tract signals at transcript 3'-ends, and through its nuclease domain cleaves downstream, so cooperates with the terminator to dictate TTEs. Therefore, distinct to the two independent termination mechanisms, of Rho- and intrinsic terminator-dependent respectively, in bacteria, a two-in-one termination mode, that the trans-action factor aCPSF1 and the intrinsic terminator U-tract cis-element cooperatively determines high TTEs, could be employed in archaea. The termination relying on aCPSF1 cleavage at transcript 3'-end resembles the eukaryotic RNAP II termination mode (*Yue et al., 2020*). However, unlike eukaryotic RNAP II termination, where multiple protein factors are involved (*Figure 7* and *Dominski, 2010*; *Eaton et al., 2018*; *Hill et al., 2019*; *Kuehner et al., 2011*; *Porrua et al., 2016*), aCPSF1 may be the only trans-action factor for triggering archaeal termination, by its N-terminal KH domain specifically recognizing and binding to the terminator U-tract and its β-CASP domain implementing transcript 3'-end cleavage, as depicted in *Figure 7*. Given the wide distribution of the terminator U-tract as well as the aCPSF1 orthologs among archaea, and two aCPSF1 orthologs from distant relatives exhibited the same cooperation with the terminator U-tracts, the two-in-one termination mode in dictating high TTEs may be widely employed by archaeal phyla.

## aCPSF1 specifically recognizes the U-tract terminator signal at transcript 3′-end

Earlier studies have identified oligo(dT) stretches in non-template DNA strands usually upstream the TTSs of the coding and non-coding RNAs in archaeal species (*Brown et al., 1989*); therefore, the TTS proximal U-rich sequences were assumed as the terminator signal that triggers intrinsic termination events in archaea analog to that in bacteria, which depend on a similar U-rich stretch but with a preceding hairpin structure (*Brown et al., 1989*; *Maier and Marchfelder, 2019*; *Thomm et al., 1993*). The terminator U-tracts, usually comprised of four to seven consecutive uridines, are determined to be essential for efficient termination by both in vitro transcription and an in vivo reporter systems through inserting the tested terminators +10 nt downstream from the transcription start sites in *T. kodakarensis*, (*Santangelo et al., 2009*; *Santangelo and Reeve, 2006*; *Thomm et al., 1993*). Nevertheless, the in vitro assay may not distinguish the terminator U-tract merely caused pausing or also dismantled TEC from the DNA template. Moreover, inserting terminator closely downstream of the transcription start site in the in vivo reporter system may introduce artificial assay biases and may not accurately convey the virtual transcription termination action, as abortive transcriptions usually occur at the transcription initiation phase caused by RNAP reiterative initiation (*Blombach et al., 2019*; *Smollett et al., 2017*).

This work, through comprehensive bioinformatic analysis and experimental confirmations demonstrated that neither aCPSF1 nor the terminator U-tract is dispensable and that the two cooperatively control the effective and hierarchical transcription termination of *M. maripaludis*. Genome-level Term-seq analysis indicated that both the terminator U4-tracts and the aCPSF1 dependency are linearly correlated with the TTEs of 961 identified TTSs of TUs (*Figures 1 and 2*) and 76.6% (736/961) TUs were found to have both U4-tract terminator- and aCPSF1-dependencies (*Figure 1—figure supplement 4A*), and also revealed that terminators containing more U4-tracts have higher aCPSF1 dependency (*Figure 2C and D*). Further, the in vitro biochemical assays demonstrated that aCPSF1 binds more strongly to the terminators containing more U-tracts, and two U4-tracts may represent the minimum specific sequence for efficient binding of aCPSF1 (*Figures 3 and 4*, *Figure 3—figure supplements 1 and 2*, 3 and *Figure 4—figure supplements 1 and 2*, 3). Meanwhile, the terminator reporter assays verified the cooperation of aCPSF1 and the terminator U-tracts in dictating high TTEs in vivo (*Figure 5*). Therefore, we conclude that the termination factor aCPSF1 through specifically binding to the terminator U-tracts dictates the genome-wide TTEs in *M. maripaludis*. Given the wide distributions of both aCPSF1 and the terminator U-tracts in archaeal species (*Berkemer et al., 2020*; *Dar et al., 2016a*; *Yue et al., 2020*), the cooperative mode of aCPSF1 and terminator U-tracts may be widely employed by archaeal phyla. Additionally, aCPSF1 was also found to be involved in the termination of 15% (144/961) of TUs with weak terminator signals lacking the U4-tract in *M. maripaludis* (*Figure 1—figure supplement 4A*) through a remained unknown mechanism. Therefore, by collaborating with the intrinsic terminator U-tracts for high TTEs and providing a guarantee for weak terminators with no U-tract, aCPSF1-dependent termination could serve as the principal termination mechanism in archaea.

## Both the KH domains in N-terminal and the conserved C-terminal sequence are indispensable for aCPSF1 in implementing transcription termination

Since identified (*Siomi et al., 1993*), the KH domain, either as single or multiple copies, has been widely found in proteins frequently involved in transcriptional, posttranscriptional, and translational regulation, and equip the proteins in targeting specific RNA sequences (*Beuth et al., 2005*; *Biswas et al., 2019*; *Gibson et al., 1993*; *Wasmuth et al., 2014*; *Wong et al., 2013*). KH domains are restricted present in the aCPSF1 orthologs among the archaeal β-CASP family ribonucleases (*Clouet-d'Orval et al., 2015*; *Dominski et al., 2013*; *Mir-Montazeri et al., 2011*; *Nishida et al., 2010*; *Silva et al., 2011*), suggesting that aCPSF1 could target more specific RNA sequences than other β-CASP proteins, such as aRNase J and aCPSF2, which only contain MβL and β-CASP domains. This work determined that the KH domains of *M. maripaludis* aCPSF1 exhibit a binding specificity to native terminator U-rich sequences preceding TTSs in comparable to the whole-length protein (*Figures 3, 4 and 6A*). Complementation experiments demonstrated that the KH domains are essential for aCPSF1 to mediate transcription termination (*Figure 6B–D* and *Figure 6—figure supplement 1*). The necessity

of U-tract recognition by KH domains of aCPSF1 in dictating TTEs was further confirmed by in vivo terminator reporter assays (*Figure 5*). Therefore, the in vitro and in vivo experiments prove that the two N-terminal KH domains equip aCPSF1 with the ability to specifically bind the terminator U-tract sequences, and that such binding is vital for aCPSF1 to function in transcription termination in vivo.

Additionally, it indicates that the aCPSF1 orthologs occur as dimers in solution, and the C-terminal 13 residues with a conserved motif sequence was proposed mediating the dimerization (*Mir-Montazeri et al., 2011*; *Phung et al., 2013*; *Silva et al., 2011*). The C-terminal 12 residue (ΔCter) deleted variant of *Pyrococcus abyssi* aCPSF1 was reported to impair the dimerization, RNA binding, and the endonucleolytic activity in vitro (*Phung et al., 2013*). This study, found that disrupting the dimerization of aCPSF1 caused its defect in transcription termination (*Figure 6—figure supplement 3*). Therefore, neither the N-terminal KH domains nor the conserved C-terminal sequence are dispensable for aCPSF1 to implement transcription termination.

## aCPSF1-dependent archaeal transcription termination may represent a simplified eukaryotic RNAP II termination model

Based on the findings in this work, we propose that the multi-functional domain aCPSF1 may be the only protein factor to trigger archaeal transcription termination through performing both terminator signal recognition and transcript 3′-end cleavage (*Figure 7*). First, the N-terminal KH domains of aCPSF1 could recognize the intrinsic U-tract terminator signals at nascent transcript 3′-ends, and the direct association of aCPSF1 with aRNAP provides the spatial vicinity for the recognition. Upon specific binding, the nuclease module of aCPSF1 could perform endoribonucleolytic cleavage downstream to produce the transcript 3′-end, which has been determined to be necessary for transcription termination (*Yue et al., 2020*) and the U-tract mediated hierarchical TTEs in vivo as well (*Figure 5B*). Therefore, the termination factor aCPSF1 could implement termination by the sole protein factor, itself, in archaea, through the N terminal KH domains recognizing and specifically binding to the terminator U-tracts and the nuclease domain cleaving at the transcript 3′-ends.

Although the transcript 3′-end cleavage in aCPSF1-dependent termination resembles with that of the eukaryotic RNAP II termination mode, wherein an aCPSF1 homolog, the yeast Ysh1 or human CPSF73, in the multi-subunits 3′-end processing machinery executes the 3′-end cleavage (*Dominski, 2010*; *Eaton et al., 2018*; *Hill et al., 2019*; *Kuehner et al., 2011*; *Porrua et al., 2016*). However, the eukaryotic CPSF73/Ysh1 lacks the KH domains in archaeal aCPSF1, but requires the interacting subunits in the CPF/CPSF complex and the accessory cleavage factors IA and IB (CFIA and CFIB) to recognize the poly(A) terminator signal and the flanking U-rich sequences to facilitate the 3′-end cleavage (*Hill et al., 2019*; *Porrua and Libri, 2015*). Very few homologs of the eukaryotic 3′-end processing complex (*Casañal et al., 2017*; *Shi et al., 2009*) and termination factors (*Baejen et al., 2017*; *Eaton et al., 2018*; *Grzechnik et al., 2015*; *Larochelle et al., 2018*) have been identified in archaea. Therefore, the archaeal multiple domains aCPSF1 may play the equivalent role of the eukaryotic multi-subunit 3′-end processing/termination machinery in the function of transcription termination, while the transcript 3′-end cleavage mode and the U-rich sequence recognition could be the evolutionary relic retained in the eukaryotic termination mechanism, namely that the aCPSF1-dependent archaeal termination mechanism reported here could be a simplified and evolutionary predecessor of the eukaryotic transcription termination machinery.

In conclusion, this work reports that the intrinsic terminator U-tract cis-element and the trans-acting termination factor aCPSF1 cooperatively dictate the effective and hierarchical TTE in archaea at the genome-wide level. The two-in-one mechanism, but not two independent termination systems, may be the principal termination mechanism of archaea. Although resembling the eukaryotic RNAP II termination mode, archaea may employ a simplified termination strategy, as aCPSF1, a single protein factor, could initiate archaeal transcription termination through (i) specifically binding of the N-terminal KH domains to the terminator U-tract and (ii) cleavage at the transcripts 3′-end by the nuclease domain. Therefore, the archaeal transcription termination mode might represent the archetype of the eukaryotic RNAP II transcription termination.

# Materials and methods

## Strains, plasmids, and culture conditions

Strains and plasmids used in this study are listed in *Supplementary file 4a*. *M. maripaludis* S2 and its derivatives were grown in pre-reduced McF medium under a gas phase of $N_2/CO_2$ (80:20) at 37°C and 22°C as previously described (*Sarmiento et al., 2011*), and 1.5% agar was used in the solid medium. Neomycin (1.0 mg/ml) and puromycin (2.5 µg/ml) were used for genetic modification selections unless indicated otherwise. *E. coli* DH5α, BL21(DE3)pLysS and BW25113 were grown at 37 °C in Luria-Bertani (LB) broth and supplemented with ampicillin (100 µg/ml), streptomycin (50 µg/ml) or kanamycin (50 µg/ml) when needed.

## Term-Seq data analysis

TTS analysis was based on the Term-seq data obtained in previous study based on two parallel replicates (*Yue et al., 2020*). To minimize the possibility of sites derived from stale RNA processing or degradation products, TTSs were all assigned within 200 nt downstream the stop codon of a gene to maximumly enrich the authentic TTSs near the gene 3′-ends and only sites that appeared in the two replicates with high coverage were recorded. The primary TTSs were identified as described previously (*Yue et al., 2020*). Other TTSs except the primary TTSs were assigned as the secondary TTSs (*Supplementary file 1*) by fitting the following three criteria: (i) within 200 nt downstream the stop codon of a gene; (ii) > 1.1 read ratio of –1 site (predicted TTS) to +1 site (one nt downstream TTS); (iii) read-counts of –1 site minus +1 site > 5. Motif sequence logos were created using WebLogo (http://weblogo.threeplusone.com/) developed by Schneider and Stephens (*Crooks et al., 2004*). The lengths of 3′UTR were defined as the distance between the stop codons to the TTSs and listed in *Supplementary file 1*. Number of U4 tracts in the range from –31 site (30 nt upstream the TTS) to –1 site (identified TTS) of each primary TTSs was searched and listed in *Supplementary file 1*. Additionally, the TTSs of non-coding RNAs in intergenic regions were manually annotated using the similar criteria that for gene TTSs except for the item of within 200 nt downstream the stop codon of gene.

Transcription termination efficacy (TTE) for transcript with Term-seq identified TTS was calculated by the formula as follow,

$$\text{Transcription termination efficacy }(\text{TTE}) = 1 - S2\left[+2\right]/S2\left[-2\right]$$

where S2[+2] and S2[–2] indicate the read-counts of +2 (2 nt downstream TTS) and –2 site (2 nt upstream TTS) in strain S2, respectively. Next, TTS Quadruplet Read Ratio (TQRR) was developed to define the transcription termination dependency on aCPSF1 using the following equation:

$$\text{TQRR} = \frac{S2\left[+2\right]/S2\left[-2\right]}{\triangledown aCPSF1\left[+2\right]/\triangledown aCPSF1\left[-2\right]}$$

where S2[+2] and S2[–2] indicate the read-counts of +2 and –2 site in strain S2, respectively; $\triangledown$aCPSF1[+2] and $\triangledown$aCPSF1[–2] indicate the read-counts of +2 and –2 site in strain $\triangledown$aCPSF1, respectively. Based on the criteria, those with TQRR <1 were defined as aCPSF1-dependent terminators.

## Rapid amplification of cDNA 3′-Ends (3′RACE)

3′RACE was performed as described previously (*Zhang et al., 2009*). Total RNA was extracted from the cells cultured to the exponential phase, and a total of 20 µg were ligated with 50 pmol 3′adaptor Linker (5′-rAppCTGTAGGCACCATCAAT–NH₂-3′; NEB) via a 16 h incubation at 16 °C using 20 U T4 RNA ligase (Ambion). Isopropanol precipitation was performed for the recovery of the 3′ linker-ligated RNA ; one aliquot of 2 µg recovered 3′ linker-ligated RNA was mixed with 100 pmol 3′R-RT-P (5′-ATTGATGGTGCCTACAG-3′, complementary to the 3′ RNA linker) by incubation at 65 °C for 10 min and on ice for 2 min. Then, the reverse transcription (RT) reaction was performed using 200 U SuperScript III reverse transcriptase (Invitrogen). After RT, nested PCR was conducted using primers (*Supplementary file 4b*) that target regions of <200 nt upstream the termination codons to obtain gene-specific products. Specific PCR products were excised from 2% agarose gel and cloned into pMD19-T (TaKaRa) and then sequenced. The 3′-end of a TU is defined as the nucleotide linked to the 3′RACE-linker.

## Protein purification of aCPSF1 and its N-terminal KH domains

The heterogeneous expression and purification of *Mmp*-aCPSF1 in *E. coli* have been described in detail previously (*Yue et al., 2020*). For purification of the aCPSF1 KH domains, the 1–147 amino acids of *Mmp*-aCPSF1 including the N-terminal two KH domains was inserted into pET28a using the ClonExpress MultiS One Step Cloning Kit. The constructed plasmid was then transformed into *E. coli* BL21(DE3)pLysS to heterogeneously express a His-tagged aCPSF1-KH domain recombinant protein. After a 16 hr induction at 22 °C, the cells were harvested, and the protein was purified through a His-Trap HP column and followingly a Q-Trap HP column as described previously (*Zheng et al., 2017*). Purified proteins were detected by SDS-PAGE, and the protein concentration was determined using a BCA protein assay kit (Thermo Scientific).

## RNA electrophoretic mobility shift assay (rEMSA)

The rEMSA assay was performed as described previously with some modifications (*Li et al., 2019*). In Brief, an RNA binding assay was performed in a 10 µl reaction mixture containing binding buffer (20 mM HEPES-KOH, pH 7.5, 1 mM $MgCl_2$, 150 mM NaCl, and 5% of glycerol), 0.5 nM 3′-biotin-labeled RNA substrate, and increasing amounts of purified recombinant proteins of the catalytic inactive aCPSF1 and its N-terminal KH domains (aCPSF1-KH). After incubation at 25 °C for 30 min, the reaction mixtures were loaded onto a 6% polyacrylamide gel and electrophoresed under 100 V for 50 min in 0.5× TBE running buffer. The free RNA and RNA-protein complexes in gels were transferred onto a nylon membrane and cross-linked using GS Gene Linker UV Chamber (Bio-Rad Laboratories, Hercules, CA). The nylon membrane was incubated with 20 µg/mL of proteinase K (Ambion) at 55 °C for 2 hr, then a Chemiluminescent Nucleic Acid Detection Module kit (Thermo Scientific) was used to detect chemiluminescence by exposure on a Tanon-5200 Multi instrument (Tanon Science & Technology Co. Ltd., Shanghai, China).

## Surface plasmon resonance (SPR) assay

A surface plasmon resonance (SPR) assay was performed to determine the binding affinities of aCPSF1 to poly-U-tract containing RNA on a BIAcore 8 K instrument (GE Healthcare) as described previously (*Zhang et al., 2017*) with minor modifications. A streptavidin-coated sensor chip SA (Series S Sensor chip SA, GE Healthcare) was first conditioned with three injections (10 µl min$^{-1}$) of buffer containing 1 M of NaCl and 50 mM of NaOH until a stable baseline was obtained. The 3′-biotinylated RNA was then diluted to 200 nM in binding buffer (20 mM HEPES-KOH, pH 7.5, 1 mM $MgCl_2$, 150 mM NaCl, 5% of glycerol, and 0.05% Tween 20) and immobilized in flow cell two at a flow rate of 10 µl min$^{-1}$ for 5 min. NaCl (500 mM) was then injected at 5 µl min$^{-1}$ to remove unbound RNA molecules until the response units (RU) reached a stable state. *Mmp*-aCPSF1 was twofold serially diluted from 1000–0 nM with binding buffer and continuously injected into flow cells with RNA immobilized and the control flow cell one without RNA of the sensor chip simultaneously at room temperature. The signal of flow cells with RNA was subtracted from that of flow cell one to eliminate nonspecific interactions. BSA was included as a negative control. The sensorgrams were analyzed using Biacore Insight Evaluation Software (version 1.0.5.11069, GE Healthcare).

## The RNase footprint assay

The RNase footprint assay was performed by following the procedure described in *Nilsen, 2014*. The 36 nt RNA carrying the two U-tract terminator sequence of T0204 was 5′-end labeled with [γ-$^{32}$P] ATP (PerkinElmer) using T4 polynucleotide kinase (Thermo Scientific). The appropriate digestion condition of RNase I in the reaction was first established by titrate the RNase usage for digesting 100 nM RNA (25,000 cpm/µl) at 25 °C for 8 min. Then, the rEMSA assay of the purified aCPSF1 binding to the [γ-$^{32}$P] ATP labelled RNA was performed in a 9 µl reaction mixture by following the similar procedure as described above for the 3′biotin-labeled RNAs, and the binding reaction without aCPSF1 was used as control. After the rEMSA reactions were incubated at 25 °C for 20 min, 0.3 U RNase I and 1 µl yeast tRNA of 10 mg/ml were added and further incubated for 8 min. The reaction products were then mixed with formamide-containing dye and analyzed on 20% sequencing urea-PAGE. A nucleotide ladder was generated by alkaline hydrolysis of the labeled RNA substrate. The urea-PAGE gels were analyzed by autoradiography with X-ray film.

## Construction of the ΔKH-aCPSF1 and aCPSF1-ΔC13 complementary strain

The *Mmp*-aCPSF1 (*MMP0694*) expression depleted strain (▽*aCPSF1*) was constructed using a *TetR-tetO* repressor-operator system as described previously (*Yue et al., 2020*). To obtain an aCPSF1-ΔKH complementary strain (Com(ΔKH)), the plasmid pMEV2-*aCPSF1*-ΔKH was first constructed. To obtain this plasmid, the fragment *aCPSF1*ΔKH-pMEV2 was amplified from the plasmid pMEV2-*aCPSF1* using primers pMEV2-*aCPSF1*-ΔKH-F/R listed in **Supplementary file 1b**. Through the Gibson assembly via ClonExpress Ultra One Step Cloning Kit (Vazyme), the fragment *aCPSF1*ΔKH-pMEV2 was then circled to form the plasmid pMEV2-*aCPSF1*-ΔKH. The constructed plasmid was transformed into ▽*aCPSF1* via the PEG-mediated transformation approach to produce the complementary strains (**Tumbula et al., 1994**). The *Mmp*-aCPSF1-ΔC13 complementary strain (Com(ΔC13)) was constructed in a similar way except amplifying the fragment *aCPSF1*-ΔC13-pMEV2 from the plasmid pMEV2-*aCPSF1* using primers pMEV2-*aCPSF1*-ΔC13-F/R listed in **Supplementary file 1b**.

## Western blot assay

Western blot was performed to determine the cellular *Mmp*-aCPSF1, aCPSF1-ΔKH, or aCPSF1-ΔC13 protein abundances in various genetic modified strains as described previously (*Yue et al., 2020*). A polyclonal rabbit antiserum against the purified *Mmp*-aCPSF1 protein was raised by MBL International Corporation, respectively. The mid-exponential cells of *M. maripaludis* were harvested and resuspended in a lysis buffer [50 mM Tris-HCl (pH 7.5), 150 mM NaCl, 10 (w/v) glycerol, 0.05% (v/v) NP-40], and lysed by sonication. The cell lysate was centrifuged and proteins in the supernatant were separated on 12% SDS-PAGE and then transferred to a nitrocellulose membrane. The antisera of anti-*Mmp*-aCPSF1 (1: 10,000) were diluted and used respectively, and a horseradish peroxidase (HRP)-linked secondary conjugate at 1:5000 dilutions was used for immunoreaction with the anti-*Mmp*-aCPSF1 antiserum. Immune-active bands were visualized by an Amersham ECL Prime Western blot detection reagent (GE Healthcare).

## Construction of the in vivo terminator reporter system

To test the TTE of each terminator in vivo, the terminator reporter system was constructed by putting the tested terminator between the upstream luciferase and downstream mCherry gene. The promoter of *MMP1697* was used to transcribe this transcript because this promoter has similar transcription ability in strains S2 and ▽*aCPSF1*. To obtain this reporter system, we first synthesis the DNA fragment P1697-luciferase-tested terminator-mCherry-T0952, in which P1697 and T0952 are the promoter of *MMP1697* and the terminator of *MMP0952*; luciferase and mCherry are the two codon-optimized fluorescent reporter genes. For the homologous recombination of the synthesized fragment to the *M. maripaludis* genome, the genome region of the intergenic region (IGR) between *MMP0852* and *MMP0853* was chosen as this IGR is long enough to avoid polar effect to downstream genes. Then, to get the homologous arms, the DNA fragment from the upstream of *MMP0852* to the downstream of *MMP0853* was PCR amplified from the S2 genomic DNA using primers *MMP0852up-F* and *MMP0853down-R*, and inserted to pMD19-T (TAKARA) to obtain pMD19-T-MMP0852/0853 (**Supplementary file 4a**). The synthesized DNA fragment of P1697-luciferase-Reporter terminator-mCherry-T0952 and the resistance gene were inserted into the IGR between *MMP0852* and *MMP0853* via ClonExpress Ultra One Step Cloning Kit to get the plasmid pReporter-Ter. Finally, this reporter system was transformed into *M. maripaludis* S2 and its derivatives via the PEG-mediated transformation approach to produce the reporter system strains (**Tumbula et al., 1994**). Different terminator sequence fragments were changed through Gibson assembly technique by the primer listed in **Supplementary file 4b**.

## Quantitative RT-PCR for determining the transcription abundance of reporter genes

Total RNA was extracted from the mid-exponential cells as described previously (*Yue et al., 2020*). 500 ng Total RNA were digested and reverse transcribed using ReverTra Ace qPCR RT Master Mix with gDNA Remover (Toyobo) according to the supplier's instructions and used for qPCR amplification with the corresponding primers (**Supplementary file 4b**). Amplifications were performed with a Mastercycler ep realplex2 (Eppendorf AG, Hamburg, Germany). To estimate copy numbers of the

luciferase and mCherry mRNA, a standard curve of their genes was generated by quantitative PCR using 10-fold serially diluted PCR product as the template. The number of copies of mCherry transcript per luciferase transcript copies is shown. All measurements were performed on triplicate samples and repeated at least three times.

## Acknowledgements

The authors thank Prof. William B Whitman at the University of Georgia providing strain *Methanococcus maripaludis* S2 and plasmids pIJA03 and pMEV2. The authors also thank Zheng Fan for surface plasmon resonance assay and Jingfang Liu for recombinant protein identification by MS, and LetPub (https://www.letpub.com) for linguistic assistance on the manuscript preparation.

## Additional information

### Funding

| Funder | Grant reference number | Author |
|---|---|---|
| National Key R&D Program of China | 2019YFA0905500 | Xiuzhu Dong |
| National Key R&D Program of China | 2020YFA0906800 | Jie Li |
| National Natural Science Foundation of China | 91751203 | Xiuzhu Dong |
| National Natural Science Foundation of China | 32070061 | Xiuzhu Dong |

The funders had no role in study design, data collection and interpretation, or the decision to submit the work for publication.

### Author contributions

Jie Li, Conceptualization, Data curation, Formal analysis, Funding acquisition, Investigation, Methodology, Project administration, Supervision, Validation, Visualization, Writing – original draft, Writing – review and editing; Lei Yue, Data curation, Formal analysis, Methodology, Project administration, Resources, Supervision, Validation; Zhihua Li, Wenting Zhang, Data curation; Bing Zhang, Formal analysis, Resources, Software; Fangqing Zhao, Software; Xiuzhu Dong, Conceptualization, Funding acquisition, Supervision, Validation, Writing – original draft, Writing – review and editing

### Author ORCIDs

Jie Li http://orcid.org/0000-0002-2773-6806
Xiuzhu Dong http://orcid.org/0000-0002-6926-5459

### Decision letter and Author response

Decision letter https://doi.org/10.7554/eLife.70464.sa1
Author response https://doi.org/10.7554/eLife.70464.sa2

## Additional files

### Supplementary files

• Supplementary file 1. Termseq identified TTSs and the read ratios of 1 to TTS 1 in *M. maripaludis* S2 and ▽aCPSF1.

• Supplementary file 2. The TTEs and TQRRs of primary TTSs.

• Supplementary file 3. The TTEs of TTSs for noncoding RNAs in the wild type S2 and ▽aCPSF1 mutant.

• Supplementary file 4. Includes *Supplementary file 4a-4e*.

• Transparent reporting form

## Data availability

Term-seq data have been deposited at the NCBI GEO Submission (GSE141346). All data generated or analysed during this study are included in the manuscript and supporting files including supplementary Figures, Tables and Datasets.

The following dataset was generated:

| Author(s) | Year | Dataset title | Dataset URL | Database and Identifier |
|---|---|---|---|---|
| Yue L Li J | 2019 | RNA-seq and Term-seq of Methanococcus maripaludis wild-type strain S2 and aCPSF1 depletion strain ∇aCPSF1 | https://www.ncbi.nlm.nih.gov/geo/query/acc.cgi?acc=GSE141346 | NCBI Gene Expression Omnibus, GSE141346 |

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
