## [Editor Report]

The process of termination in Archael species is poorly defined despite a high relation to eukaryotes and a shared homology of termination factors. In this study, the authors defined key features that drive termination to include an upstream uridine track that is bound by the CPSF ribonuclease through KH RNA binding domains not present in the CSPF counterparts. This work provides fundamental mechanistic insight into the conserved manner of termination in the archael species.

---

## [Decision Letter]

**Decision letter after peer review:**

Thank you for submitting your article "aCPSF1 in synergy with terminator U-tract dictates archaeal transcription termination efficiency via the KH domains recognizing U-tract" for consideration by *eLife*. Your article has been reviewed by 3 peer reviewers, one of whom is a member of our Board of Reviewing Editors, and the evaluation has been overseen by James Manley as the Senior Editor. The following individuals involved in review of your submission have agreed to reveal their identity: Béatrice Clouet-d'Orval (Reviewer #2); Bin Tian (Reviewer #3).

Essential revisions:

1) Provide quantitative analysis of gel shift assays to calculate affinity constants.

2) Conduct more in depth in vitro experiments – in particular foot print assays – in order to provide additional mechanistic insight beyond previous published work from this research team

*Reviewer #1 (Recommendations for the authors):*

1) The use of a stop codon in the term-seq pipeline seems limiting – although it is not clear to me what level of noncoding RNA and/or how well these are annotated in this species is. The authors should consider methods to analyze Term-seq that falls through this filter to identify any similar/different motifs that might be present in noncoding genes.

2) The authors criteria of TTUs appearing in biological replicates with significant coverage needs to be defined in the text. What p value cutoffs were used? What defines significant coverage?

3) I disagree with the authors statement of how the TTS quadruplet was defined. The authors claim a 'step-wise manner' of termination is the likely cause of this signature, but this is confusing. Isn't it simpler that termination is imprecise and occurs most often within these four nucleotides, but the actual termination site is subject to natural variation? To interpret this to mean that termination is occurring in steps seems incorrect.

4) I understand what the others are trying to claim by calculating efficiency of termination but this doesn't seem to be the correct use of this word. Perhaps accuracy or frequency?

5) the authors claim that high TTE have an over-representation of U tracks and low TTE have under-representation but it seems that this can be quantified in some manner and the significance of this enrichment also could be provided.

6) the authors do a great job correlating data from CPSF deletion strains and how often a U-track is found but I am having trouble understanding the conclusion that CPSF 'backs up' termination when there isn't a strong U track. How is this concluded?

*Reviewer #2 (Recommendations for the authors):*

As general comments, the experiments and their analyses are well-conducted. However, additional experiments are necessary to go deeper in the understanding of how aCPSF1 specifically recognized PolyU-tract signals. First, a more precise picture of how aCPSF1 binds to PolyU-tract could be obtained by quantifying the gel retardation experiments (EMSA) or/and SPR binding assays. This will give affinity constants for aCPSF1 proteins (WT or ∆KH domain) binding to poly-U tract substrates. In addition, footprint assays could allow determining how aCPSF1 sits on PolyU-tract. Finally, the authors should consider including experiments with aCPSF1 invalidated for its C-terminal region (last 16-AA) which has been shown to be important for the dimerization of aCPSF1. It is important to decipher if aCPSF1 is acting as a monomer or a dimer. Footprint experiments should also be carried with this mutant.

In addition, the writing of the manuscript should be improved. The authors should revise the manuscript by choosing a more appropriate vocabulary and a more synthetic writing, using shorter sentences and avoiding repetition. This will facilitate the reading and highlight the important key points.

Below are the detailed specific comments by section of the manuscript.

Vocabulary:

• The term "efficiency" is not appropriate. It should be replaced by "effectiveness" or "efficacy".

• What the authors meant by "heterogeneous aCPSF1 orthologs" (lane 420) ?

• Two-in-one termination mode? To my understanding, it is a one mode of termination that implicated a cis-element and aCPSF1

• The title should be shortened by removing "via the KH domain recognizing U-tract"

Results/Figures:

By reinvestigating their Term-seq data, the authors state they want to evaluate the role of aCPSF1 on "transcription termination efficiency". In my opinion, using the term "efficiency" is not appropriate. Indeed, the authors do not define the efficiency of termination by analysing the Term-seq data but rather its effectiveness: they aim at defining the degree to which transcription termination is successful in producing the transcript 3' end, and not how quickly it can be done. Therefore, the term "efficiency" should be replaced by "effectiveness" or "efficacy".

Why is there such heterogeneity at the 3' end from +2 to -2? Is the heterogeneity comes from a random cleavage by aCPSF1 and/or to additional exonucleolytic activities?

The Term-seq data analysis specifically identified TTS of mRNAs by searching the 200-nt sequence located downstream a stop codon. What about transcription termination of noncoding RNAs? Is aCPSF1/U-tract dependent transcription termination specific for mRNAs? or can it drive transcription termination of ncRNAs? Is there Poly-U stretch at the 3' end of ncRNAs ? Nonetheless, in the description of the results, the authors should clearly specify that their results point to the 3' end of mRNAs.

The authors need to define their ∆aCPSF1 mutant more clearly by indicating that it still exhibits 20% of aCPSF1 molecules (Yue et al., 2020). From Figure 6B, when expressing the ∆KH-aCPSF1, the quantity of wild type aCPSF1 seems even higher. They should take these into account when discussing their results.

While the authors defined the TQRR as TTS quadruplet ratio, it is almost never expressed as such. Somehow, it becomes percentages with no explanation from the authors. In Figure 2B and 2D, while the y-axis is labelled as "TTS quadruplet read ratios" (please, note that the 't' of 'ratios' is missing), it exhibits percentages as values. This makes the results hard to follow. The same comment stands for the TA Ratios when measuring transcript abundance.

EMSA results showing binding of aCPSF1 to poly-U are only qualitative and not quantitative. Quantification of retardation gels (EMSA) and/or SPR binding assays is necessary to define the affinity constants of each substrate and each protein (WT or ∆KH domain). Furthermore, the quantitative analysis of the SPR experiment could allow determining the association and dissociation constants. To go further on the understanding of the binding of aCPSF1 on U-tract , footprint experiments could be performed. This will be helpful in understanding the mechanism of action of aCPSF1.

aCPSF1 has been shown to be a dimeric enzyme. This dimerization relies on the last 16aa of aCPSF1 which are conserved across species. The authors should test if this motif is important for the function of aCPSF1 to terminate transcription in their assays and for RNA binding.

In Yue et al. 2020, an association of aCPSF1 and the RNAPol was determined through co-immunoprecipitation. It would be interesting to know if the ΔKH aCPSF1 mutant still interacts with RNAPol.

In the manuscript, the use of the term synergy to define the action of aCPSF1 and the polyU-tract seems inadequate. While the two seems indispensable for transcription termination, there is no evidence that they work in synergy. Synergy is an interaction that produces an overall effect greater than the sum of individual effects of any of them. At most, the authors can say that aCPSF1 and the polyU-tract cooperate in terminating transcription.

Finally, as a general comment, on several occasions, the conclusions of the experiments are missing and let to the interpretation of the readers. For example, see Lanes 367 to 372. The authors should also revise the title of the Figures that are not informative.

Discussion

The Discussion section is too long and difficult to follow. A more synthetic writing will facilitate the reading and highlight the important key points. For example the paragraph Lanes 527 to 546 could be synthetically summarized in few short sentences.

The authors seem to imply that aCPSF1 is acting alone without additional factors (Lanes 456 to 462). It might be due to the phrasing that can be improved. If not, from the manuscript, it is not clear how they can make such a statement.

In the model presented in Figure 7, the authors do not propose 5'-3' Exo activities (from aCPSF1 or other β-CASP ribonucleases) as being involved in the removal of the cleaved RNAs attached to the RNAP as it is proposed in the case of the eukaryal RNApolII. This was part of their previous model in Yue et al. 2020? Do the authors have evidence that it is not the case?

The authors should not forget that aCPSF1 could have additional roles in the cell. Indeed, a study in H. volcanii identified Hvo-aCPSF1 as a protein partner of NcsA that is involved in the formation of thiolated tRNA (Lys)UUU together with homologues of ubiquitin-proteasome (Chavarria et al. 2014).

*Reviewer #3 (Recommendations for the authors):*

In this manuscript by Li et al. examine U-rich motifs enriched for the transcript end sites in archaea M. marpaludis and analyze the role of aCPSF1 and its KH domains in binding these U-rich motifs. Their data indicate aCPSF1 binding to U-rich motifs is necessary for efficient 3' end definition of transcripts. Overall this work is well carried out. However, there several key issues the authors need to address before the paper can be accepted.

1. The conclusion that aCPSF1 functions as a back-up termination is not supported by their data. As far as I can tell, back-up termination should happen at non-primary sites, which have a lower frequency of U quadruplex. It is not clear how aCPSF1 functions at those sites.

2. The authors appear to indicate that aCPSF1 is the sole factor for 3' end cleavage of archaeal transcripts. But this is not supported by the data. Their data indicate both U-rich motif and aCPSF1 are necessary for cleavage. No data are shown to indicate that these two alone are sufficient for 3' end cleavage.

3. The U-rich motif (U quadruplex) was recently reported in their NAR paper (Yue et al.). There appears to be limited additional information on motifs in this work. It would be useful to readers to know if U-rich motifs are the only type for 3' end cleavage. The authors may want to examine motifs beyond single nucleotide models (which is what they are doing in this work). For example, are there any k-mer enrichments besides U quadruplex?

[Editors' note: further revisions were suggested prior to acceptance, as described below.]

Thank you for resubmitting your article "aCPSF1 cooperates with terminator U-tract to dictate archaeal transcription termination efficacy" for consideration by *eLife*. Your article has been reviewed by 2 peer reviewers, and the evaluation has been overseen by a Reviewing Editor and James Manley as the Senior Editor. The following individuals involved in review of your submission have agreed to reveal their identity: Béatrice Clouet-d'Orval (Reviewer #2); Bin Tian (Reviewer #3).

The reviewers have discussed their reviews with one another, and the Reviewing Editor has drafted this to help you prepare a revised submission. Overall, the Reviewers were satisfied with the revised manuscript but we ask that comments raised by Reviewer #2 be addressed. No experiments are suggested and the comments are only meant to improve clarity in the text and figures.

*Reviewer #2 (Recommendations for the authors):*

The authors have improved the manuscript by addressing the main criticisms raised in the first round of review. However, one key issue described below remains to be addressed.

The authors should show on Figure 3 and 4 the binding curves that allow determining Kd values. It is not clear how these values are calculated and the number of experimental replicates is not mentioned. Comparison of the binding curves on the same graph would highlight the differences in affinity between the different substrates. Moreover, in view of the values of the affinity constants, it is unlikely to obtain values (µM) with a standard deviation of up to three digits after the decimal point. The Kd values should be revised. Finally, I do not think it is possible to determine Kd if the RNA is not 100% bound to the highest protein concentration. This is the case of the substrates: T0204-18nt (Figure 4B) , T0229-18nt (Figure 4C) and T02046-DU5 -DU4 (Fig4D).

*Reviewer #3 (Recommendations for the authors):*

I am satisfied with the authors' response to my comments. I have no other issues to raise.

---

## [Author Response]

Essential revisions:1) Provide quantitative analysis of gel shift assays to calculate affinity constants.2) Conduct more in depth in vitro experiments – in particular foot print assays – in order to provide additional mechanistic insight beyond previous published work from this research team

Based on the comments and corrections from the three reviewers and the required essential revisions, we have thoroughly revised the manuscript in the following aspects:

1. The affinity constants were provided to quantify the analysis of gel shift assays (the revised Figures 3, 4, and 6A);

2. Footprint assay of aCPSF1 binding to the native U-tract terminator was provided, which added a further mechanistic insight upon gel shift assays (Figure 4—figure supplement 2 in reversion);

3. Supplemented the data of transcription termination of the non-coding RNAs and found that it also depends on aCPSF1 recognizing the U-tract terminator. These data include the terminator motif, the correlation of TTEs and TQRRs, and the transcription readthrough evidences detected by the stand-specific transcriptomic mapping profiles and the northern blot assays for non-coding RNAs, and have been supplemented in Figure 2—figure supplement 2 and 3；

4. Supplemented the data verifying that the N-terminal KH domains of the aCPSF1 are not essential to the interaction with RNA polymerase (Figure 6—figure supplement 2);

5. Supplemented the experimental data that aCPSF1 dimerization is essential for its role in transcription termination (Figure 6—figure supplement 3), which is evaluated by truncating the C-terminal 13 residues that are required for aCPSF1dimerization.

Reviewer #1 (Recommendations for the authors):1) The use of a stop codon in the term-seq pipeline seems limiting – although it is not clear to me what level of noncoding RNA and/or how well these are annotated in this species is. The authors should consider methods to analyze Term-seq that falls through this filter to identify any similar/different motifs that might be present in noncoding genes.

Thank you for the valuable advice. As you say, the term-seq pipeline filtered out majority of the noncoding RNA TTSs. Therefore, the TTSs of non-coding RNAs in *M. maripaludis* (Supplementary File 3 in revision) were intensively and manually analyzed based on the Term-seq data by following the same criteria for the coding genes except for the item of within 200 nt downstream the stop codon of a gene. The noncoding RNAs of *M. maripaludis* (Supplementary File 3) were annotated based on our previous transcriptomic data, the published literatures and the KEGG databases. Noteworthily, similar U-tract motif, and similar cooperative trend of TTEs and aCPSF1 dependency were found present in noncoding RNAs (Figure 2—figure supplement 2). Additionally, the aCPSF1 dependency for the termination of two representative noncoding RNAs was verified by the prolonged transcript 3′-ends detected by both the transcriptomic sequencing and the northern blot assay (Figure 2—figure supplement 3) in ▽*aCPSF1*. These supplemented data suggest that the noncoding RNAs of *M. maripaludis* could employ the similar termination mechanism as that of the genes, i.e. depending on both aCPSF1and the U-tract terminator. The related description has been supplemented in the revised manuscript (Lines 242-252).

2) The authors criteria of TTUs appearing in biological replicates with significant coverage needs to be defined in the text. What p value cutoffs were used? What defines significant coverage?

The previous inaccurate description of “significant” coverage has been revised as “high coverage” in the results and methods (Lines 146 and 646), which is based on the TTSs identification criteria as the sites present in the two replicates and with high coverage and additionally fitting the following three criteria: (i) within 200 nt downstream the stop codon; (ii) >1.1 read ratio of -1 site (predicted TTS) to +1 site (1 nt downstream TTS) and (iii) read-counts of -1 site minus +1 site >5.

3) I disagree with the authors statement of how the TTS quadruplet was defined. The authors claim a 'step-wise manner' of termination is the likely cause of this signature, but this is confusing. Isn't it simpler that termination is imprecise and occurs most often within these four nucleotides, but the actual termination site is subject to natural variation? To interpret this to mean that termination is occurring in steps seems incorrect.

Thank you for the suggestion and we agree with your opinion. The previous inappropriate description has been revised as “a dramatic decreasing pattern was observed in the mapping reads at four nucleotides (nts) between sites +2 and −2 flanking TTS (−1 nt) in the majority of the primary TTSs (Figure 1A and Figure 1—figure supplement 3). This indicates that transcription appears to be terminated most frequently at the four nucleotides, … (Lines 162-166). The related descriptions in figure legends of Figure 1A and Figure 1—figure supplement 3 have been revised as well (Lines 979, 982, and 1012).

4) I understand what the others are trying to claim by calculating efficiency of termination but this doesn't seem to be the correct use of this word. Perhaps accuracy or frequency?

Thank you for the suggestion. Transcription termination efficiency (TTE) has been revised to be “Transcription termination efficacy” throughout the revised manuscript.

5) The authors claim that high TTE have an over-representation of U tracks and low TTE have under-representation but it seems that this can be quantified in some manner and the significance of this enrichment also could be provided.

According to your suggestion, we used Wilcox test analyzing significant of the U tract number between the three groups with different TTEs, which detected the P values between Groups I and II, I and III, II and III to be 3.4e-12, 2.22e-16 and 2.1e-5, respectively. The relevant descriptions have been added in the text (Lines 178-180) and the legend of Figure 1C (Lines 991-993).

6) the authors do a great job correlating data from CPSF deletion strains and how often a U-track is found but I am having trouble understanding the conclusion that CPSF 'backs up' termination when there isn't a strong U track. How is this concluded?

Combination of the comments of you and the reviewer 3, this inappropriate conclusion has been removed in the revised manuscript.

Reviewer #2 (Recommendations for the authors):As general comments, the experiments and their analyses are well-conducted. However, additional experiments are necessary to go deeper in the understanding of how aCPSF1 specifically recognized PolyU-tract signals. First, a more precise picture of how aCPSF1 binds to PolyU-tract could be obtained by quantifying the gel retardation experiments (EMSA) or/and SPR binding assays. This will give affinity constants for aCPSF1 proteins (WT or ∆KH domain) binding to poly-U tract substrates. In addition, footprint assays could allow determining how aCPSF1 sits on PolyU-tract. Finally, the authors should consider including experiments with aCPSF1 invalidated for its C-terminal region (last 16-AA) which has been shown to be important for the dimerization of aCPSF1. It is important to decipher if aCPSF1 is acting as a monomer or a dimer. Footprint experiments should also be carried with this mutant.In addition, the writing of the manuscript should be improved. The authors should revise the manuscript by choosing a more appropriate vocabulary and a more synthetic writing, using shorter sentences and avoiding repetition. This will facilitate the reading and highlight the important key points.

According to your comments and suggestions, we have supplemented the following experimental data in the revision as (i) the aCPSF1-U rich RNA binding association contents (K_d_) have been quantified for the binding assays; (ii) truncation of the C-terminal 13 residues (C13) that are essential for aCPSF1 dimerization leads to the loss of the in vivo termination function of aCPSF1, demonstrating that the dimerization of aCPSF1 is essential to its routine function; (iii) Footprint assay also determined that aCPSF1 binds to the U-tract region of the tested terminator. In addition, English language and grammar have revised by a professional English language editing company. We hope the language and grammar have been improved.

Below are the detailed specific comments by section of the manuscript.Vocabulary:• The term "efficiency" is not appropriate. It should be replaced by "effectiveness" or "efficacy".

Thank you for the suggestion. "Efficiency" has been revised as "efficacy" throughout the manuscript.

• What the authors meant by "heterogeneous aCPSF1 orthologs" (lane 420)?

"heterogeneous aCPSF1 orthologs" intended to indicate that aCPSF1 orthologs are from remote relatives like *Loki*-aCPSF1 from Asgard archaea and *Csy*-aCPSF1 from Thaumarchaeota. To avoid ambiguity, “heterogeneous” has been removed from the related sentence. (Line 440)

• Two-in-one termination mode? To my understanding, it is a one mode of termination that implicated a cis-element and aCPSF1.

“Two-in-one termination mode” is proposed to disprove the “two independent termination modes” proposed by Sanders et al., 2020; Wenck and Santangelo, 2020, which proposed that analogous to bacteria, there could be two independent termination modes in archaea: the U-rich terminator dependent and trans-action factor aCPSF1 mediated. Whereas, our study found that the two elements in fact cooperatively determine genome-wide transcription termination in archaea. This viewpoint has also been described in the discussion part (Lines 465-471).

• The title should be shortened by removing "via the KH domain recognizing U-tract"

The title has been shortened according to the suggestion.

Results/Figures:By reinvestigating their Term-seq data, the authors state they want to evaluate the role of aCPSF1 on "transcription termination efficiency". In my opinion, using the term "efficiency" is not appropriate. Indeed, the authors do not define the efficiency of termination by analysing the Term-seq data but rather its effectiveness: they aim at defining the degree to which transcription termination is successful in producing the transcript 3' end, and not how quickly it can be done. Therefore, the term "efficiency" should be replaced by "effectiveness" or "efficacy".

Thank you for the suggestion. "efficiency" has been revised to be "efficacy" throughout the manuscript.

Why is there such heterogeneity at the 3' end from +2 to -2? Is the heterogeneity comes from a random cleavage by aCPSF1 and/or to additional exonucleolytic activities?

About the 3' end heterogeneity, we agree about the reviewer 1’s viewpoint that the termination is imprecise and occurs most often within these four nucleotides, and the actual termination site for each RNA is subject to natural variation with high frequency in these four nucleotides of the Term-seq obtained. Therefore, the related description has been revised as “This indicates that transcription appears to be terminated most frequently at the four nucleotides, which was therefore defined as the TTS quadruplet.” (Lines 165-166). Given that the specific sequence recognized by aCPSF1 is polyU, the TTS heterogeneity may probably come from an unprecise cleavage by aCPSF1 downstream the polyU sequence.

The Term-seq data analysis specifically identified TTS of mRNAs by searching the 200-nt sequence located downstream a stop codon. What about transcription termination of noncoding RNAs? Is aCPSF1/U-tract dependent transcription termination specific for mRNAs? or can it drive transcription termination of ncRNAs? Is there Poly-U stretch at the 3' end of ncRNAs ? Nonetheless, in the description of the results, the authors should clearly specify that their results point to the 3' end of mRNAs.

Thank you for the valuable advice. As you say, the term-seq pipeline filtered out majority of the noncoding RNA TTSs. Thorough intensively manual analysis using the same criteria for the coding genes except for the item of within 200 nt downstream the stop codon of a gene, we found that the noncoding RNAs of *M. maripaludis* (the newly supplemented Dataset S3), which were annotated based on our previous transcriptomic data, the published literatures and the KEGG databases, could employ the similar mechanism as that of the genes, i.e. dependent on the cooperation of the terminator factor aCPSF1 and the terminator U-tracts for dictating TTEs (Figure 2—figure supplement 2 and 3). Similar U-tract motif was found present in noncoding RNAs, and similar cooperative trend of TTEs and aCPSF1 dependency also occurs (Figure 2—figure supplement 2). Additionally, the aCPSF1 dependency for the termination of two representative noncoding RNAs was verified by the prolonged transcript 3’-ends detected by both the transcriptomic sequencing and the northern blot assay (Figure 2—figure supplement 3) in ▽*aCPSF1*. The related description has been supplemented in the revised manuscript (Lines 242-252), and the results point to the 3' end of coding RNAs (mRNAs) were also indicated in the revised version (Lines 157, 207, 209, 218, 252, and 1031).

The authors need to define their ∆aCPSF1 mutant more clearly by indicating that it still exhibits 20% of aCPSF1 molecules (Yue et al., 2020). From Figure 6B, when expressing the ∆KH-aCPSF1, the quantity of wild type aCPSF1 seems even higher. They should take these into account when discussing their results.

Thank you for the point. The description of 20% residual protein abundance of aCPSF1 in ▽*aCPSF1* mutant has been supplemented in the revised manuscript (Line 199). Yes, a little higher intensity (1.5-fold quantified by Image J) of the WT aCPSF1 was detected in the Com(∆KH) strain, and this has been discussed as suggested: “Although the truncated ΔKH-aCPSF1 and an even ~1.5-fold increase of the whole-length aCPSF1 were detected in Com(ΔKH) (Figure 6B), the same growth defect as ▽*aCPSF1* was found for Com(ΔKH), in contrast to the strain Com(WT), which was complemented with the whole-length wild-type *aCPSF1* gene and restored the growth defect of ▽*aCPSF1* (Figure 6C).” (Lines 370-374). However, a little higher WT aCPSF1 and the complementary ∆KH-aCPSF1 were not capable of recovering the growth retardation and the transcription termination defects (Figure 6C and 6D).

While the authors defined the TQRR as TTS quadruplet ratio, it is almost never expressed as such. Somehow, it becomes percentages with no explanation from the authors. In Figure 2B and 2D, while the y-axis is labelled as "TTS quadruplet read ratios" (please, note that the 't' of 'ratios' is missing), it exhibits percentages as values. This makes the results hard to follow. The same comment stands for the TA Ratios when measuring transcript abundance.

Thank you for reminding, and “Percentage” has been revised to “ratio” throughout the revised manuscript and the missing 't' of 'ratios' in Figure 2B and 2D has been added.

EMSA results showing binding of aCPSF1 to poly-U are only qualitative and not quantitative. Quantification of retardation gels (EMSA) and/or SPR binding assays is necessary to define the affinity constants of each substrate and each protein (WT or ∆KH domain). Furthermore, the quantitative analysis of the SPR experiment could allow determining the association and dissociation constants. To go further on the understanding of the binding of aCPSF1 on U-tract , footprint experiments could be performed. This will be helpful in understanding the mechanism of action of aCPSF1.

Thank you for the suggestions. Equilibrium dissociation constants (Kd) were calculated by the binding curves that were generated through quantifying the unbound and bound substrates in Figures 3, 4, and 6A. The calculated Kd values are indicated in the respective Figures.

Additionally, the footprint experiments to detect aCPSF1 binding to U-tract RNA was performed, which confirmed the specific binding of aCPSF1 to the U-rich region. The related description has been supplemented in the revised manuscript (Lines 302-305, 730-744, and 1111-1116).

aCPSF1 has been shown to be a dimeric enzyme. This dimerization relies on the last 16aa of aCPSF1 which are conserved across species. The authors should test if this motif is important for the function of aCPSF1 to terminate transcription in their assays and for RNA binding.

By truncating the C-terminal 13 residues in *mmp*-aCPSF1 (aCPSF1-ΔC13), the role of the dimerization of aCPSF1 in its in vivo termination function was evaluated. Complementation of aCPSF1-ΔC13 failed to restore the transcription termination and growth defects of ▽aCPSF1, demonstrating that the dimerization of aCPSF1 is essential to its routine function in transcription termination. The related results and discussion have been supplemented in the revised manuscript (Lines 401-415, 566-577, and 1176-1187).

In Yue et al. 2020, an association of aCPSF1 and the RNAPol was determined through co-immunoprecipitation. It would be interesting to know if the ΔKH aCPSF1 mutant still interacts with RNAPol.

Interaction of ΔKH-aCPSF1 with RNAP was evaluated using the similar method as described previously (Yue et al., 2020). A gel filtration coupled western-blot experiment has been performed on Com(ΔKH) strain, which detected the co-occurrence of aCPSF1 and its KH domain deletion mutant with RNAP subunit RpoD, indicating that the ΔKH-aCPSF1 mutant still interacts with the RNAP. These results have been supplemented in the revised manuscript (Lines 383-385 and 1168-1175).

In the manuscript, the use of the term synergy to define the action of aCPSF1 and the polyU-tract seems inadequate. While the two seems indispensable for transcription termination, there is no evidence that they work in synergy. Synergy is an interaction that produces an overall effect greater than the sum of individual effects of any of them. At most, the authors can say that aCPSF1 and the polyU-tract cooperate in terminating transcription.

Thank you for the suggestion. “synergy” has been revised as “cooperation” throughout the – revised manuscript.

Finally, as a general comment, on several occasions, the conclusions of the experiments are missing and let to the interpretation of the readers. For example, see Lanes 367 to 372. The authors should also revise the title of the Figures that are not informative.

Thank you for the suggestions. The missing conclusions of the related experiments have been added (Lines 385-390). The non-informative titles of Figures have been revised.

DiscussionThe Discussion section is too long and difficult to follow. A more synthetic writing will facilitate the reading and highlight the important key points. For example the paragraph Lanes 527 to 546 could be synthetically summarized in few short sentences.

Thank you for the suggestion. The related paragraph and the Discussion section have been abbreviated to be more synthetic.

The authors seem to imply that aCPSF1 is acting alone without additional factors (Lanes 456 to 462). It might be due to the phrasing that can be improved. If not, from the manuscript, it is not clear how they can make such a statement.

To avoid ambiguity, the related description has been revised as “…, the working mode of archaeal termination is also noteworthy as aCPSF1 may be the only trans-action factor in triggering termination, by …” (Lines 475-479) in the revision. We mean that aCPSF1 is the only protein factor for the U-tract terminator recognition and transcript 3' end cleavage in archaeal termination process, which is distinct from its eukaryotic homologous CPSF73, which only performs the 3' end cleavage but dependent other interacted proteins in the multiple-subunits CPSF complex in eukaryotic termination process.

In the model presented in Figure 7, the authors do not propose 5'-3' Exo activities (from aCPSF1 or other β-CASP ribonucleases) as being involved in the removal of the cleaved RNAs attached to the RNAP as it is proposed in the case of the eukaryal RNApolII. This was part of their previous model in Yue et al. 2020? Do the authors have evidence that it is not the case?

Thank you for the point. 5'-3' Exo activities were proposed and added in Figure 7, though not experimentally proved.

The authors should not forget that aCPSF1 could have additional roles in the cell. Indeed, a study in H. volcanii identified Hvo-aCPSF1 as a protein partner of NcsA that is involved in the formation of thiolated tRNA (Lys)UUU together with homologues of ubiquitin-proteasome (Chavarria et al. 2014).

As you have pointed, aCPSF1 could have additional roles in the cell. In addition of regulating the NcsA-mediated thiol modification of wobble uridine, we previously found approximate two-fold longer bulk mRNA half-life in 22°C-cultured ▽*aCPSF1* than in the wild-type (8.4 min vs 4.7 min, Figure 1*C* in Yue et al. 2020), suggesting the potential role of aCPSF1 functioning as an archaeal house-keeping ribonuclease in mRNA turnover.

Reviewer #3 (Recommendations for the authors):In this manuscript by Li et al. examine U-rich motifs enriched for the transcript end sites in archaea M. marpaludis and analyze the role of aCPSF1 and its KH domains in binding these U-rich motifs. Their data indicate aCPSF1 binding to U-rich motifs is necessary for efficient 3' end definition of transcripts. Overall this work is well carried out. However, there several key issues the authors need to address before the paper can be accepted.1. The conclusion that aCPSF1 functions as a back-up termination is not supported by their data. As far as I can tell, back-up termination should happen at non-primary sites, which have a lower frequency of U quadruplex. It is not clear how aCPSF1 functions at those sites.

Combination of the comments of you and the reviewer 1, this related inappropriate conclusion has been removed in the revised manuscript.

2. The authors appear to indicate that aCPSF1 is the sole factor for 3' end cleavage of archaeal transcripts. But this is not supported by the data. Their data indicate both U-rich motif and aCPSF1 are necessary for cleavage. No data are shown to indicate that these two alone are sufficient for 3' end cleavage.

The obscure related description has been revised as “the only trans-action factor”. (Line 36 in the Abstract and Lines 476-477)

3. The U-rich motif (U quadruplex) was recently reported in their NAR paper (Yue et al.). There appears to be limited additional information on motifs in this work. It would be useful to readers to know if U-rich motifs are the only type for 3' end cleavage. The authors may want to examine motifs beyond single nucleotide models (which is what they are doing in this work). For example, are there any k-mer enrichments besides U quadruplex?

According to your advice, we searched RNA sequences proximal to TTSs; however, no other motif in addition the U-rich sequence is found.

[Editors' note: further revisions were suggested prior to acceptance, as described below.]

The reviewers have discussed their reviews with one another, and the Reviewing Editor has drafted this to help you prepare a revised submission. Overall, the Reviewers were satisfied with the revised manuscript but we ask that comments raised by Reviewer #2 be addressed. No experiments are suggested and the comments are only meant to improve clarity in the text and figures.

Based on the comments and corrections from Reviewer #2, we have revised the manuscript in the following aspects:

1. The binding curves for the rEMSA gels in Figure 3 and 4 have been supplemented in Figure 3—figure supplement 2 and Figure 4—figure supplement 2 as required, and the methods of calculating the Kd values and the number of experimental replicates have all supplemented in the related figure legends;

2. The Discussion section has been further concentrated to focus on three key points and we hope the writing could be more synthetic and readable;

3. The aCPSF1 in Figure 7 has been sketched as a dimer as suggested;

4. The related reference has been cited accordingly (lines 526 and 811-813 in the revised manuscript).

Reviewer #2 (Recommendations for the authors):The authors have improved the manuscript by addressing the main criticisms raised in the first round of review. However, one key issue described below remains to be addressed.The authors should show on Figure 3 and 4 the binding curves that allow determining Kd values. It is not clear how these values are calculated and the number of experimental replicates is not mentioned. Comparison of the binding curves on the same graph would highlight the differences in affinity between the different substrates. Moreover, in view of the values of the affinity constants, it is unlikely to obtain values (µM) with a standard deviation of up to three digits after the decimal point. The Kd values should be revised. Finally, I do not think it is possible to determine Kd if the RNA is not 100% bound to the highest protein concentration. This is the case of the substrates: T0204-18nt (Figure 4B) , T0229-18nt (Figure 4C) and T02046-DU5 -DU4 (Fig4D).

The binding curves for the rEMSA gels in Figure 3 and 4 have been supplemented in Figure 3—figure supplement 2 and Figure 4—figure supplement 2. The Kd values have been revised accordingly by showing one digit after decimal point. Moreover, the Kd values for the substrates mentioned (T0204-18nt, T0229-18nt and T02046-DU5 -DU4) were calculated through the same methods for other RNA substrates shown in Figure 3 and 4. The binding curve for each indicated RNA was obtained through quantifying the shifted RNAs (%) vs. the protein concentrations and the Kd values were calculated by the software Prism using the nonlinear regression method. The Kd values and the standard deviations are calculated from three independent binding assays, which has been indicated in the figure legends of Figure 3, 4, Figure 3—figure supplement 2, and Figure 4—figure supplement 2.